# Dynamic Modeling of Patients, Modalities and Tasks via Multi-modal Multi-task Mixture of Experts

**Chenwei Wu,*** **Zitao Shuai,*** **Zhengxu Tang,*** **Luning Wang & Liyue Shen** [†]
Department of Electrical and Computer Engineering, University of Michigan, liyues@umich.edu

## Abstract

Multi-modal multi-task learning holds significant promise in tackling complex diagnostic tasks and many significant medical imaging problems. It fulfills the needs in real-world diagnosis protocol to leverage information from different data sources and simultaneously perform mutually informative tasks. However, medical imaging domains introduce two key challenges: dynamic modality fusion and modality-task dependence. The quality and amount of task-related information from different modalities could vary significantly across patient samples, due to biological and demographic factors. Traditional fusion methods apply fixed combination strategies that fail to capture this dynamic relationship, potentially underutilizing modalities that carry stronger diagnostic signals for specific patients. Additionally, different clinical tasks may require dynamic feature selection and combination from various modalities, a phenomenon we term "modality-task dependence." To address these issues, we propose $M^4oE$, a novel **M**ulti-modal **M**ulti-task **M**ixture of Experts framework for precise **M**edical diagnosis. $M^4oE$ comprises Modality-Specific (MSoE) modules and a Modality-shared Modality-Task MoE (MToE) module. With collaboration from both modules, our model dynamically decomposes and learns distinct and shared information from different modalities and achieves dynamic fusion. MToE provides a joint probability model of modalities and tasks by using experts as a link and encourages experts to learn modality-task dependence via conditional mutual information loss. By doing so, $M^4oE$ offers sample and population-level interpretability of modality contributions. We evaluate $M^4oE$ on four public multi-modal medical benchmark datasets for solving two important medical diagnostic problems including breast cancer screening and retinal disease diagnosis. Results demonstrate our method's superiority over state-of-the-art methods under different metrics of classification and segmentation tasks like Accuracy, AUROC, AUPRC, and DICE.[1]

## 1 Introduction

Multi-modal and multi-task learning holds strong potential for tackling many significant and challenging medical imaging problems (Tu et al., 2024; Khara et al., 2024b; Cai et al., 2024; He et al., 2021), as it mimics real-world clinical protocols of disease diagnosis, where clinicians leverage all available data sources and simultaneously perform multiple assessments that are inherently interrelated and informative of each other (Stahlschmidt et al., 2022; Acosta et al., 2022). For example, radiologists combine full-field digital mammography (FFDM) containing high-resolution details and 2D synthesized (2DS) images indicating high-contrast tissue structures for breast density prediction and cancer risk screening (Brown et al., 2023); Ophthalmologists integrate color fundus photographs with optical coherence tomography (OCT) that uses near-infrared light and interferometry for diabetic retinopathy diagnosis and pan-retinal leakage detection (Korot et al., 2021). All these applications request a model that can understand the complementary information from these

---

[*]These authors contribute equally to this work.

[†]Corresponding Author

[1]Our code could be found in M4oE Official Implementation.

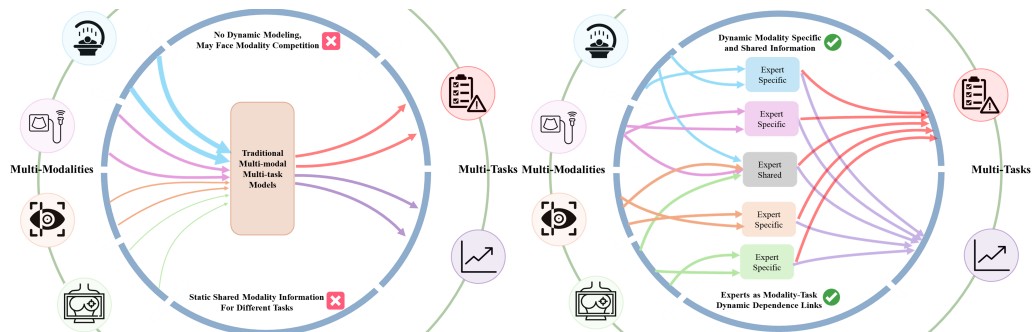

Figure 1: Introduction: Traditional multi-modal multi-task modeling vs M$^4$oE in medical imaging

different input modalities for predicting various tasks, drawing a lot of research attention (Yang, 2024; Zhao et al., 2022).

Modeling the complex cross-modal and cross-task relationships of medical imaging modalities is a non-trivial challenge. Different modalities can maintain both modality-specific and modality-shared information: for example, while MRI and PET can both be used for Alzheimer's disease analysis, MRI allows for higher soft-tissue contrast and PET for metabolic activity visuals (Pichler et al., 2008; Zhang et al., 2023). Such information should be inherently dynamic across patient samples due to different patient subgroups and environmental variances. This is analogous to the well-known dynamic fusion of infrared and RGB images in the natural domain. In RGB-Infared fusion, the relative importance of each modality naturally varies with environmental conditions - RGB provides better information in daylight while Infrared excels in low-light conditions. In breast cancer screening, parenchymal tissues in younger women often appear similar and confuse with tumor tissues in FFDM images, whereas it can be visualized more clearly by 2DS, due to reduced tissue overlaps (Khara et al., 2024a). For older patients, where calcifications are a key sign of early cancer, FFDMs are more reliable in showing these small spots (Aujero et al., 2017; Brown et al., 2023). These insights from practical medical applications inspire us to raise the first research question: *How can we capture the patient-dynamic modality-shared and modality-specific information in multimodal data fusion*?

Existing multi-modal fusion methods for medical images can be mainly categorized into early fusion (Jiang et al., 2019), late fusion (Donnelly et al., 2024; Weisman et al., 2022), and information-theory-based methods (Zhang et al., 2024; Chen et al., 2023b). However, most of these approaches are developed based on the assumption that different modalities contain fixed information distribution across different samples, which rarely holds true in the real world. We conjecture that ignoring such reality leads to the problem of "Modality Competition,"(Huang et al., 2022) where only a single modality dominates the joint latent space, suppressing distinct domain-specific features that should commonly exist in all modalities. This can result in severely compromised predictions when crucial disease information is only present in an underrepresented modality. We hypothesize that models should be able to dynamically integrate effective information from the respective modalities while not losing generalizable information shared in different modalities. That is, dynamically learning **sample-dynamic modality-specific** and **modality-shared** information.

Another challenge comes with the various interrelated target tasks. Different diagnostic tasks require a task-specific way to select and fuse features from different imaging modalities, depending on the modality-task relevance. For instance, in a comprehensive breast cancer screening utilizing top and side views of both FFDM and 2DS images (4 modalities) (Khara et al., 2024a), density assessment relies more on FFDM due to its ability to provide higher-resolution details, while the disease detection task, particularly for calcified cancers and lesions in dense tissue, benefits significantly from 2DS with better visibility scores (Aujero et al., 2017; Destounis et al., 2020). This implies the second important research question: *How do we model the dynamic modality dependence on different tasks?* Current multi-task learning methods either aim to learn a shared set of features that generalize across different tasks (Swamy et al., 2024) or focus on separating task gradients or task modules for a single modality (Chen et al., 2023c), ignoring the dynamic nature of task-relevant modality-shared information. We hypothesize that models should be able to dynamically leverage the most relevant features from each modality to fuse for each specific task while not compromising the diversity of

shared multi-modal features generalizable across different tasks. In other words, the goal is to learn the **task-specific** and **task-shared** manifolds in the multi-modal latent space.

In short, as illustrated in Figure 11, effective fusion of multi-modalities should be both sample-dynamic and task-dependent. Motivated by this, in this work, we propose $\mathbf{M^4oE}$, a new **M**ulti-modal **M**ulti-task **M**ixture of Experts (MoE) framework for **M**edical imaging, dynamically linking modalities, samples and tasks via soft mixture of expert (Puigcerver et al., 2023) networks. $M^4oE$ consists of two main modules: Modality-Specific MoE (MSoE) and Modality-Shared Modality-Task MoE (MToE). The combination of modality-shared and modality-specific experts offers a promising solution to the modality competition problem, achieving sample dynamic fusion. By leveraging the input-dependent nature of MoE, our proposed model dynamically learns distinct and shared information from different modalities. Besides, MToE explicitly models the modality-task dependence by learning a joint distribution of modalities, experts, and tasks. By optimizing conditional mutual information loss, we encourage experts to learn diverse patterns of task-specific modality-shared information. Our experiments and analysis show that $M^4oE$ can be well generalized to solve different multimodal medical problems in various applications. More importantly, we show that our proposed MToE module can be flexibly combined with existing multi-modal fusion strategies to further improve performance.

Our main contributions can be summarized as follows:

1) We propose a novel framework for multi-modal multi-task learning, achieving both sample-dynamic modality fusion and dynamic modality-task dependence modeling. It provides sample-level and dataset-level interpretation of modality contribution to different prediction tasks.

2) Our proposed MToE module provides a simple yet effective way to jointly model and interpret modality, expert, and task dependency. We devise a new conditional mutual information loss to encourage experts to dynamically learn diverse patterns of task-dependent modality-shared information for different target tasks, respectively.

3) Extensive experiments are conducted on four multi-modal medical benchmark datasets to address two important multi-modal and multi-task medical problems: breast cancer screening and retinal diagnosis. Results show that $M^4oE$ outperforms the baselines from both medical and natural image domains, demonstrating the effectiveness and generalization of our approach.

## 2    RELATED WORK

**Multi-modal learning in Medical Imaging.** Multi-modal learning in medical imaging focuses on improving prediction and mimics the multi-modal nature of clinical expert decision-making by fusing disparate data sources (Kline et al., 2022). It has wide applications in fields such as breast cancer screening (two views of FFDM and two views of 2DS) (Yala et al., 2022), retinal diagnoses (Retinal Fundus images and OCTs) (Zou et al., 2024b), and head-neck cancer survival analyses (Meng et al., 2023) (Computed Tomography(CT), Positron Emission Tomography (PET), and Magnetic resonance imaging (MRI)). Existing works can be categorized into early fusion, late fusion (usually attention-based), and intermediate fusion methods. Late fusion focuses on learning a combined or joint latent space: Donnelly et al. (2024) learns local and global features from multi-modal breast cancer images via a combination of convolutional and transformer networks; Oyelade et al. (2024) extracts both low and high-level features using twin-CNNs and uses binary optimization method to eliminate non-discriminant features in the search space. However, late fusion is prone to lose modality-specific information (Zhang et al., 2024). Early fusion concatenates different modalities into a single input and is shown to outperform late and middle fusion for brain tumor segmentation from MRI/PET/CT data (Marinov et al., 2023). For example, Xue et al. (2021) segments liver lesions by combining predictions from low and high-level feature maps in separate PET/CT decoders. In intermediate fusion, the features, respective to each modality, are joined in intermediate layers before feeding to the final prediction head (Huang et al., 2022). This type of model allows for more convenient modeling of modality-specific and shared information. Chen et al. (2019); Yao et al. (2024b); Wang et al. (2023) disentangles the features shared across modalities and those unique within each modality to address challenges like modality inconsistency and modality missingness. Marinov et al. (2023) disentangles unimodal and multi-modal features using modality-specific decoders and a multi-modal decoder. Zhang et al. (2024) uses a mutual information disentangled transformer to decompose modality information into shared and specific latent space to predict can-

cer risks with multi-modal histopathology data. Apart from prediction tasks, recent works like Yao et al. (2024a) have also extended similar fusion techniques for individualized chest X-ray generation via latent diffusion models. However, all these existing methods are either limited to single-task prediction or do not consider the dynamic dependence of modality-shared information on specific tasks, trying to learn a shared set of features for different clinical outcomes. Moreover, they all ignore the dynamic changes of modality quality in reality, which diminishes the patient-conditioned advantages of each modality, e.g., the high structural contrast from 2DS for patients with denser breasts and high-resolution details in FFDM for patients with less-dense breasts.

**Multi-task learning in Medical Imaging.** In medical imaging, multi-task learning (MTL) has wide applications as it aims to enhance the performance on interrelated tasks by leveraging shared information between them, given limited data availability (Malhotra et al., 2022). Existing medical MTL methods typically utilize hard parameter sharing methods, designing task-shared networks to capture common patterns and task-specific networks to learn individual task features (Meng et al., 2022; Ju et al., 2021). However, these methods require strong correlations among tasks (Zhang & Yang, 2018), which are not necessarily satisfied in medical scenarios. Another type of methods consider soft-parameter sharing, which utilize regularization loss terms to implicitly seperate parameters for each task (Graham et al., 2023; Zhou et al., 2021). (Shao et al., 2024) have also considered associating multi-instance medical image data with multiple tasks using attention-based fusion blocks. However, these methods assume a fixed association between tasks and input data modalities (Zhang & Yang, 2018), making them unable to dynamically capture modality-task dependencies, and thus limiting their applicability in multi-modal, multi-task medical imaging scenarios.

## 3 METHOD

### 3.1 PRELIMINARY

**Problem Formulation.** We start with the definition of multi-modal multi-task learning. Suppose we are given $m$ image modalities $M_1, \ldots, M_m$ and $p$ tasks $t_1, \ldots, t_p$, where the tasks are related but distinct. Each modality contains its own unique information, and the combination of different modalities elicits task-specific shared information. Multi-modal multi-task learning seeks to improve model performance by integrating the information from the $m$ modalities and leveraging the knowledge across the $p$ tasks. Each task $t_k$ can be viewed as a function $T_k$ that maps multi-modal image pairs $I$ to the label for each task $y_{t_k} = T_k(I)$. For each $t_k$, we have a corresponding labeled dataset $D_k$, each data point of which has $m$ paired images $I = (I_1, ..., I_m)$ and a task label $Y_{t_k}$ (e.g., paired 4 images, 2 from FFDM, 2 from 2DS, and a label of risk of breast cancer). For simplicity of analysis, we assume that each image pair $I$ is associated with a task-specific label $Y_{t_k}$ for each task $t_k$, forming the task dataset $D_k$ as $(I, Y_{t_k})$. Our approach generalizes to the more real-world scenario where each $D_k$ consists of different image pairs.

**Soft Mixture-of-Expert(MoE).** MoE enables dynamic fusion of multi-modal data and creates data-dependent flows during inference (Cao et al., 2023). Soft MoE (Puigcerver et al., 2023) replaces the top-K selection in the sparse MoE (Mustafa et al., 2022) with a softmax-based assignment in the token-expert mapping, allowing for a soft weighted combination of input tokens. Specifically, for a Soft-MoE layer with $n$ experts, given input tokens $\mathbf{X} \in \mathbb{R}^{l \times d}$, where $l$ is the number of tokens and $d$ is their dimension, the Soft MoE assigns a weighted combination of these tokens to each expert using a dispatch matrix $D \in \mathbb{R}^{l \times n}$. Each expert then projects the input tokens into output tokens, and the final output is obtained as the weighted combination of the output tokens, with the weights provided by a combine matrix $C$. The matrix $D$ and $C$ are created based on the input $\mathbf{X}$ and a learned matrix $\Phi$ of the MoE layer.

**Hypothesis.** We propose our hypotheses for medical multi-modal multi-task learning, which will be verified by our analysis experiments in the Sec. 4.3.

**Dynamic modality-shared and specific information:** Information between multi-modalities can be decomposed into modality-specific and modality-shared. This interaction should be dynamic, e.g., the high structural contrast from 2DS for patients with denser breasts and high-resolution details in FFDM for patients with less-dense breast. Ignoring this fact will cause multi-modal multi-task models to focus on optimizing parameters of a specific modality while neglecting the others, aka 'Modality-Competition'.

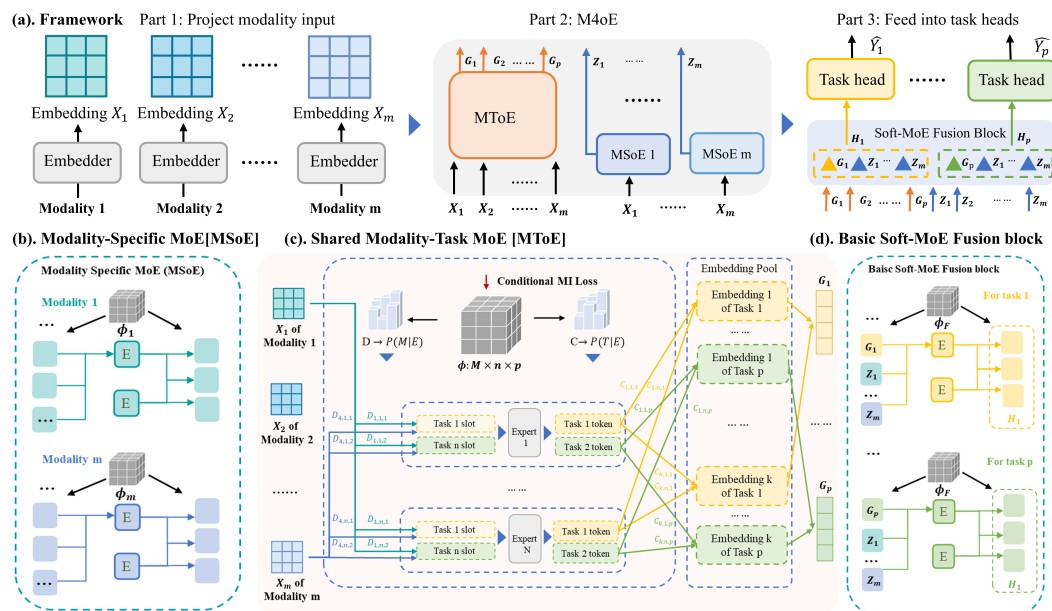

Figure 2: (a) The framework first embeds different input modalities with feature embedders, then goes through MSoE to learn and retain the modality-specific features, MToE to model shared modality information and modality-task dependence, and a final fusion block connected with task heads. (b) The MToE block creates data-dependent pathways for each multi-modal input, dispatching modality tokens to experts and combining the expert outputs to generate task-specific features. (c) Modality-specific MoEs are used to extract and retain input-dependent modality-specific features. (d). We apply a basic Soft MoE block after $M^4$oE to fuse all features.

**Dynamic modality-task dependence:** In multi-modal multi-task learning, the shared information between different modalities should be modeled differently to a given task instead of using a static shared set of features, forming a modality-task dependence.

## 3.2 OVERALL FRAMEWORK

In this section, we will demonstrate our $M^4$oE framework for multi-modal multi-task learning and illustrate how it dynamically links the input modalities to different tasks. Our $M^4$oE design comprises modality-specific feature embedders, modality-specific Soft MoEs, and shared Modality-Task Soft MoEs. We customize the MoEs into Vision Transformer blocks so that it creates a data-dependent path for each input. This enables more flexible modeling of dynamic modality dependence and dynamic multi-modal fusion in a multi-modal multi-task learning setting. Specifically, as shown in Fig. 2 a), we first utilize a feature embedder to project input modalities to embeddings $X_1, ..., X_m \in \mathbb{R}^{l \times d}$. To mitigate the modality competition phenomenon, where modality-specific information is lost for most modalities, for each modality $M_i$ we use modality-specific MoE blocks to extract and retain input-dependent modality-specific features $Z_i \in \mathbb{R}^{l \times d}$ from $X_i$. Meanwhile, we feed $X_1, \ldots, X_m$ into our proposed MToE block and obtain the task-dependent fused modality features, producing task-specific features pool $G_1, \ldots, G_p$. Finally, each task-specific feature pool from $G_1, ..., G_p$ is fused with modality-specific, task-shared feature pool $Z_1, ..., Z_m$ to form task-specific and shared pool $H_1, ..., H_p$. For each task $T_k$, $H_k$ is passed to the task heads after a basic soft MoE block for final predictions. The overall framework can be seen in Fig. 2.

The modality-specific MoE is a Soft MoE, as described in Sec. 3.1. As shown in Fig. 2 b), each MSoE $g_i$ for modality $i$ contains $n$ experts and a learnable matrix $\Phi_i \in \mathbb{R}^{d \times n}$, where $d$ represents the embedding dimension. Given input tokens $X \in \mathbb{R}^{l \times d}$, we compute the dispatch matrix $D \in \mathbb{R}^{l \times n}$ by applying softmax along the columns of $X\Phi_i$, and the combine matrix $C \in \mathbb{R}^{l \times n}$ by applying softmax along the rows of $X\Phi_i$. We pass the linearly weighted input $\tilde{X}_j$, where $\tilde{X} = DX$, to each expert $j$ and obtain the output $\tilde{Y}_j$. Finally, we apply a linear combination of $\tilde{Y}_j$ based on $C$ to obtain the final output $Y = C\tilde{Y}$ for the MSoE module.

As shown in Fig. 2 c), the MToE block connects tasks to input modalities by using experts as a link. It is distinct from the basic soft MoE in that it takes in mixed inputs from multi-modalities, directs the mixture of their information through task-specific slots, and differentiates with learnable task embeddings. The MoE layer of the MToE consists of $n$ experts and a learnable matrix $\Phi_{\text{task}} \in \mathbb{R}^{d \times n \times p}$, and each expert processes $p$ input task slots corresponding to different tasks. Given the input token-type modality features $X_1, \ldots, X_m \in \mathbb{R}^{l \times d}$, there is a total of $M = l \times m$ feature tokens. And then they are assigned to the task slots of each expert through a convex combination of $X_1, \ldots, X_m$, using dispatch weights $D \in \mathbb{R}^{M \times n \times p}$. The dispatch weight is given by applying a softmax over columns of $\mathbf{X}\Phi$, where $\mathbf{X} \in \mathbb{R}^{M \times d}$ is the concatenation of $X_1, ..., X_m$: $D_{ijk} = \frac{\exp((\mathbf{X}\Phi)_{ijk})}{\sum_{i=1}^{M} \exp((\mathbf{X}\Phi)_{ijk})}, \tilde{\mathbf{X}} = \mathbf{D}\mathbf{X}$. Each task slot $k$ of expert $j$ is processed by the expert $j$'s neural network $f_j$, to produce the output slots: $\tilde{\mathbf{Y}}_{.jk} = f_j(\tilde{\mathbf{X}}_{.jk} + TE_k)$, where $TE_k$ is the learnable task embedding of task $k$ that is shared across experts. The final output $\mathbf{Y}$ of each task $k$ is obtained by taking a convex combination of all the output slots $\tilde{\mathbf{Y}}$. This process is similar to the previous one: $C_{ijk} = \frac{\exp((\mathbf{X}\Phi)_{ijk})}{\sum_{j=1}^{n} \sum_{k=1}^{p} \exp((\mathbf{X}\Phi)_{ijk})}, \mathbf{Y}_k = \mathbf{C}_{..k} \tilde{\mathbf{Y}}_{..k}$, where $\mathbf{C}$, the combine weights, are computed by applying a softmax over the rows of $\mathbf{X}\Phi$. This facilitates the dynamic connection between different input modalities and tasks and enables probability modeling in Sec. 3.3.

### 3.3 Joint Probability Modeling of Modality, Task and Experts

To understand how each modality contributes to different tasks, we define a probability model over tasks $T$, experts $E$, and modalities $M$. We assume that when our trained network is deployed, it will receive a random task $T$ and a corresponding multi-modal image pair $I$ based on a global task distribution $P(T)$[2]. The input-conditioned and dynamic multi-expert structure of MToE makes it well-suited for jointly modeling $T$ and $M$, where the $\phi \in \mathbb{R}^{m \times n \times p}$ matrix and task slots collaboratively connect modality $M$ to expert $E$ and subsequently to task $T$.

In the MToE module, we model the probability $P(E_j|T_k)$ of selecting expert $E_j$ for task $T_k$ based on how much importance the routing network assigns expert $E_j$ to task $T_k$. For example, in sparse MoEs, if the routing network assigns 30 out of 100 image pairs from task $T_k$ to expert $E_j$, then $P(E_j|T_k) = 0.3$. In the MToE, rather than making hard assignments, we calculate these probabilities by summing the soft weights from the routing matrix $X\Phi$. Given this matrix, the conditional probability of an input $X$ can then be determined as follows: $P(E_j|T_k) = \frac{\sum_{i=1}^{m}(\mathbf{X}\Phi)_{i,j,k}}{\sum_{j=1}^{n} \sum_{k=1}^{p}(\mathbf{X}\Phi)_{i,j,k}}$, where $m$ is the number of modalities, $n$ is the number of experts, and $p$ is the number of tasks. Similarly, the conditional probability $P(M_i|T_k)$ can be represented by the frequency of how a modality $i$ is selected to contribute to a task $j$, which can be written as follows: $P(M_i|T_k) = \frac{\sum_{j=1}^{n}(\mathbf{X}\Phi)_{i,j,k}}{\sum_{i=1}^{m} \sum_{k=1}^{p}(\mathbf{X}\Phi)_{i,j,k}}$, To model how experts are assigned to each modality for different tasks, we need to model the conditional probability $P(M, E|T)$. This can be given by dispatch weights as well. Given the input $X$ and task $k$, the modality $i$ is assigned to the expert $j$ with the probability $P(M_i, E_j|T_k)$, which can be expressed by: $P(M_i, E_j|T_k) = \frac{\sum_{j=1}^{n}(\mathbf{X}\Phi)_{i,j,k}}{\sum_{i=1}^{m} \sum_{j=1}^{n} \sum_{k=1}^{p}(\mathbf{X}\Phi)_{i,j,k}}$.

Based on our hypothesis of modality-task dependence, in the multi-modal multi-task learning setting, the most important experts and the selection of input modality features should depend on the task. This allows the MToE to assign specific experts to corresponding modalities for each task. To achieve so, we could maximize the conditional mutual information between experts $E$ and modalities $M$ given $T$: $I(M; E|T) = \sum_{i=1}^{m} \sum_{j=1}^{n} \sum_{k=1}^{p} P(M_i, E_j|T_k) \log \frac{P(M_i, E_j|T_k)}{P(M_i|T_k)P(E_j|T_k)}$.

This term can be calculated based on our probabilistic modeling above in the MToE[3].

To understand this objective, we can decompose it as $I(M; E|T) = H(M|T) - H(M|E, T)$. Term $-H(M|E, T)$ refers to the entropy of modality given task and experts, a smaller $H(M|E, T)$ encourages the MToE to learn a stronger dependency of the modality on both the expert and task, and to leverage different combinations of modalities for different tasks. $H(M|T)$ refers to the entropy of modalities given tasks, where a higher term ensures modality diversity instead of repeatedly relying

---

[2]This distribution is typically derived from the dataset, as not all data points have labels for every task.

[3]In implementation, $\Phi$ is with the shape of $(m \cdot l) \times n \times p$ since each modality has $l$ tokens. For each modality, we sum up the probability of all of its tokens to marginalize probability for each Modality m.

on a small subset of modalities. Given the training objective $L_{t_k}$ of each task $t_k$, the final loss $L$ can be written as: $\mathcal{L} = \sum_{i=1}^{p} L_{t_k} - \alpha I(M; E|T)$, where $\alpha$ is the hyper-parameter that adjusts the strength of conditional mutual information loss.

## 4 EXPERIMENT

### 4.1 EXPERIMENT SETTING

**Datasets.** To test M$^4$oE's generalizability across populations and medical imaging modalities, we conduct extensive experiments over four public datasets on two use cases. **Mammogram screening:** We use EMory BrEast Imaging Dataset (EMBED) (Jeong et al., 2023) collected in the United States, The 2023 RSNA Screening Mammography Breast Cancer Detection AI Challenge Dataset (RSNA) collected in Australia (Carr et al.), and VinDr-Mammo Dataset (VinDr) collected in Vietnam (Nguyen et al., 2023). EMBED has four modalities (2 view FFDM and 2 view 2DS) and seven tasks: BI-RADS assessment (3-class), 1-5 years risk scoring (binary), and density classification(4 class). RSNA and VinDr have two modalities (2 view FFDM) and two tasks: density classification (4-class) and BI-RADS assessment (3-class). **Ophthalmology screening:** we use the Glaucoma grAding from Multi-Modality imAges (GAMMA) dataset (Wu et al., 2023) collected in China. GAMMA has two modalities (color fundus photos and OCT) and two tasks: glaucoma detection (3-class) and optical cup segmentation.

**Evaluation.** Accuracy is used for classification tasks, and DICE score is used for segmentation across 5-fold cross validation. For the M$^4$oE model reported in Table 1, we use 128 experts for the MToE module and 32 experts for each of the MSoE modules. We use Adam optimizer with learning rate of 1e-4 and stepLR scheduler with step size of 5 and gamma of 0.1. The model is trained on a batch size of 32 and a total of 100 epochs. $\alpha$ is set to 0.05. Our hardware includes 4 NVIDIA A100s and 4 L40s. Detailed mean, standard deviation, AUROC and AUPRC, are in Appendix.

**Baselines.** We mainly compare our method with four groups of SOTA methods: 1) Multi-modal single-task baselines from medical AI. 2) Multi-modal multi-task baselines from medical AI. In 1), we compare with MIRAI (Yala et al., 2022) and Asymmirai (Donnelly et al., 2024) for mammography, Eyemost (Zou et al., 2024b) and Eyestar (Wu et al., 2023) for ophthalmology. In 2), we compare with the 4 methods under multi-task setting. 3) MoE-related baselines. We compare with a multi-modal soft mixture of experts (Puigcerver et al., 2023) and the combination of MIRAI and Asymmirai with our MToE. 4) Multi-modal multi-task baselines adapted from the natural domain, including FULLER (Huang et al., 2023), AIDE (Yang et al., 2023), MModN (Swamy et al., 2024), EVIF (Geng et al., 2024). Comparison with Adapted Multimodal Soft MoE (Puigcerver et al., 2023), AdaMV-MoE (Chen et al., 2023a), and Fuse-MoE (Han et al., 2024) are included in Appendix.

### 4.2 MAIN RESULTS

**Our M$^4$oE significantly outperforms baseline methods.** As shown in Table 1, the single-task MSoE outperforms all the single-task medical AI baselines, demonstrating our design's effectiveness in retaining modality-specific information. As for multi-task settings, M$^4$oE consistently outperforms baseline methods across all tasks on all benchmarks. This highlights the effectiveness of M$^4$oE in the multi-modal, multi-task learning setting.

**Multi-tasks benefit from each other in our M$^4$oE method, unlike baselines.** In Table 1, our M$^4$oE trained in a multi-task setting shows superior performance compared to its single-task training counterpart. This indicates that tasks involved in multi-task learning contribute to shared features. We show in Sec. 4.3 that this also reduces gradient conflicts that typically hinder the learning of task-specific features. In contrast, such improvements are not seen in baseline methods. This may be because these methods fail to capture the dynamic dependence between modalities and tasks.

**Our MToE framework can generalize to different multi-modal learning backbones.** We have adapted MToE to Miral and Asymiral baselines for mammography tasks, and have adapted MToE to Eyemost and Eyestar baselines for retinal tasks. After being combined with our MToE module, the single-task oriented medical AI baselines obtain significant performance improvement, most on par with or even exceeding Multi-modal Multi-task baselines from the natural domain. This highlights MToE's flexibility and generalizability.

Table 1: Multi-modal single-task and multi-task results over four benchmarks from mammography imaging and retinal imaging. Complete 7-task results on EMBED dataset are in appendix.

| Setting | Method | EMBED | | | RSNA | | VinDR | | Method | GAMMA | |
|---|---|---|---|---|---|---|---|---|---|---|---|
| | Mammo. | Risk | Density | Birads | Density | Birads | Density | Birads | Retinal | Glau. | Seg. |
| Single -Task | Mirai | 84.0 | 82.3 | 72.5 | 76.3 | 62.3 | 86.3 | 66.1 | Eyemost | 86.0 | 87.8 |
| | Asysmirai | 79.0 | 80.2 | 69.4 | 74.1 | 60.1 | 78.9 | 62.4 | Eyestar | 85.4 | 85.9 |
| | Ours wo MSoE | 83.2 | 82.1 | 71.9 | 76.8 | 62.2 | 85.4 | 65.8 | Our wo MSoE | 86.2 | 86.2 |
| | Ours | **85.5** | **83.6** | **74.2** | **77.5** | **64.0** | **87.7** | **67.5** | Ours | **87.6** | **88.9** |
| Multi -Task | Mirai | 83.1 | 83.2 | 72.3 | 76.1 | 62.5 | 85.4 | 65.9 | EyeMost | 86.5 | 86.6 |
| | +MToE | 84.6 | 83.3 | 73.4 | 76.8 | 63.1 | 85.9 | 66.4 | +MToE | 87.2 | 88.5 |
| | Asymirai | 80.0 | 82.5 | 69.6 | 73.9 | 60.7 | 79.1 | 62.2 | Eyestar | 85.9 | 84.1 |
| | +MToE | 81.9 | 83.1 | 70.8 | 74.2 | 61.2 | 81.2 | 64.2 | +MToE | 87.5 | 86.9 |
| | EVIF | 84.7 | 83.4 | 73.6 | 77.0 | 65.9 | 88.8 | 70.7 | EVIF | 88.7 | 87.0 |
| | Fuller | 84.3 | 83.5 | 72.5 | 76.9 | 65.4 | 87.1 | 68.7 | Fuller | 87.8 | 86.2 |
| | AIDE | 84.1 | 82.9 | 73.6 | 76.8 | 66.1 | 88.2 | 69.8 | AIDE | 88.5 | 87.5 |
| | MModN | 84.8 | 83.7 | 73.9 | 77.1 | 66.4 | 89.0 | 70.4 | MModN | 89.2 | 87.2 |
| | Ours wo MToE | 84.0 | 82.8 | 73.1 | 76.7 | 64.3 | 86.9 | 67.3 | Ours wo MToE | 88.1 | 87.4 |
| | Ours | **85.9** | **84.1** | **75.1** | **77.8** | **66.7** | **89.6** | **71.8** | Ours | **90.4** | **89.7** |

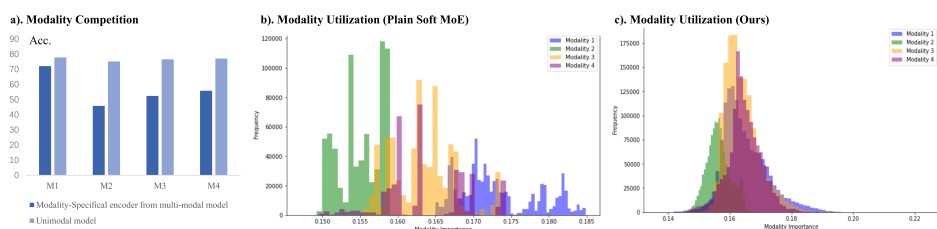

Figure 3: (a). Performance drop caused by modality competition. (b). Modality utilization of multi-modal soft MoE. (c). Modality utilization of M$^4$oE. Our method has less modality competition.

## 4.3 ANALYSIS

**Modality Utilization in Non-MoE Fusion, MoE vs M$^4$oE.** To investigate our first hypothesis, we examine how well traditional (non-MoE) multi-modal models, multi-modal MoE, and our approach utilize the four different modalities on the EMBED dataset on the cancer risk prediction task. For traditional method, we first train a multi-modal model where each modality is encoded by a modality-specific ViT encoder. The representations from each modality are then averaged and then passed through a dense fusion layer before being fed into the task heads. We also trained four unimodal models for each modality. As shown in Fig. 3(a), we compared the four modality-specific (from the multi-modal model) encoders' performance with that of their unimodal counterpart. The encoders from the multi-modal model exhibited a significant performance gap compared to the model trained solely on that modality, especially for modalities FFDM left-to-right (M2), 2DS up-to-down (M3), and 2Ds left-to-right (M4). This indicates that M2-M4 is much more poorly optimized in the multi-modal model compared to M1, i.e. losses the modality competition (Huang et al., 2022).

For the MoE-based method, it is easier to evaluate the modality utilization across experts: For modality $M_i$, we first obtain the sum of input-conditioned modality dispatch weights assigned to $M_i$'s token for a data sample and a given expert $E_j$, then aggregate this value over all experts and over all data points. In Fig. 3, we observe that when simply using a multi-modal soft MoE architecture, modality 1 dominates the expert space. In contrast, M$^4$oE alleviates this issue by learning a much more diverse modality utilization and achieves higher performance as shown in Table. 1 (Without MSoE v.s. M$^4$oE).

**Dynamic Modality Dependence.** Recall our second hypothesis that the fused features from different modalities should be dynamically dependent on different tasks. Unlike M$^4$oE, traditional multi-modal fusion models lack interpretability of modality-task dependence. To ensure a fair comparison, we introduce the synergy measure from partial information decomposition (PID) theory, which quantifies how much new task-specific information emerges when two modalities are fused.

We train synergy estimators using BATCH algorithm following Liang et al. (2024) to calculate pairwise synergy between input modalities for a given task. For this experiment, we train models on the EMBED dataset with density prediction and 1-year risk prediction tasks. We utilize the two FFDM modalities and two 2DS modalities for training, and this results in 6 pairwise synergies per task, visualized as matrices in Fig. 4. We observe that the synergy matrix for the baseline MIRAI

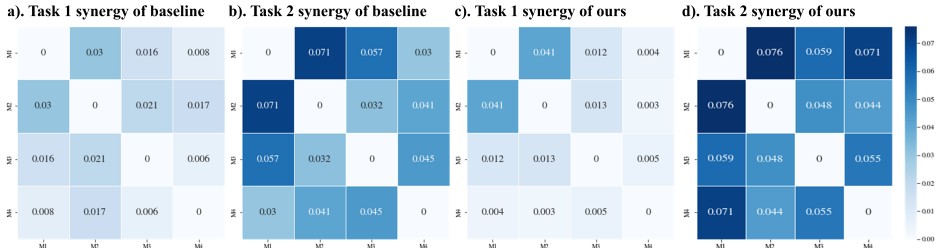

Figure 4: Visualization of modality-task dependence. We report modality-pair-wise synergies which measure how two modalities benefit from each other on contributing to the given prediction task.

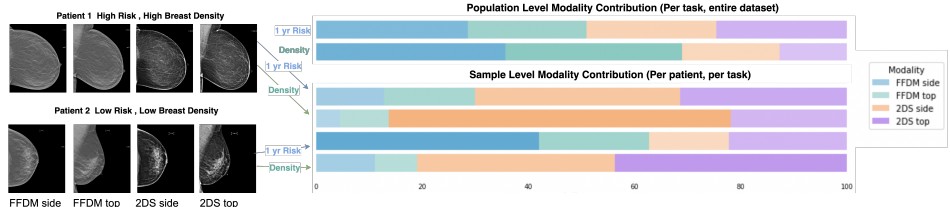

Figure 5: Visualization of modality contribution on patient level and dataset level.

method shows similar value distributions across different tasks, where shared information of two-view FFDM is prevalent (M1 and M2). While Task 1 (density) and Task 2 (risk) synergy for baseline do capture different absolute values, the highest synergy pair still only peaks for M1 and M2 (within two views of FFDM) for both tasks. However, this doesn't follow our dynamic modality-task dependence intuition. While density information is known to be mostly captured by FFDMs (Destounis et al., 2020; Zuckerman et al., 2016), prior work (Kleinknecht et al., 2020; Gastounioti et al., 2022) in mammogram screening suggests that for the cancer risk task, models should leverage useful insights from all four modalities. Our model shows a clear contrast of synergy distributions between the two tasks, and the cancer risk task's distribution derives information from more diverse information from different modalities. The benefits can be further proven by the improved results after adding MToE in Table 1 (Without MToE vs M$^4$oE). For density, our method peaks at M1-M2. For risk, we peak at not only M1-M2 synergy (within FFDM) but also M1-M4 synergy (between FFDM and 2DS). We also capture the M2-M3 and M3-M4 synergy much better than the baseline. This indicates a stronger modality-task dependency. This improvement is likely due to our method's ability to better learn task-dependent mutual information between different modalities. Further details on PID theory and BATCH algorithm can be found in Appendix A.

**Sample and Population-Level Modality Contribution Interpretability.** M$^4$oE has inherent sample and population level interpretability of modality contribution to a specific task, because the $\Phi$ matrix determines both how much weight each modality contributes to an expert and how much weight each expert contributes to a task. We use two tasks from the EMBEDS dataset to illustrate this. 1) Sample-level (local) modality contribution for one patient input: Given a task, we calculate the importance weight of each modality for each expert, and average over all experts by expert-task assignment weights. 2) Population-level (global) modality contribution: we aggregate the sample-level contribution over the entire dataset. Fig. 5 provides granular insights on how different modalities contribute to the task predictions of two different patient samples and over the entire EMBEDS dataset. We observe distinct patterns across patients and clinically sensible global patterns: FFDM contributes more to density, and all modalities should contribute to risk prediction (Destounis et al., 2020; Aujero et al., 2017; Zuckerman et al., 2016).

**Gradient Conflict Mitigation.** Gradient conflict problem is a long-standing issue in multi-task learning that may harm the model's ability to learn task-specific information (Chen et al., 2023c; Zhou et al., 2024). This issue is also present in multi-modal scenarios, as shown in Fig 6. We observe that M$^4$oE also helps mitigate this issue. We train a ViT-based multi-task late fusion model using four modalities in mammography, predicting density and 1-year risk. We collect the gradients from the last layer of the fusion block during back-propagation for both task losses in each step of an epoch. As shown in Fig 6, we calculate the cosine similarities between the gradients of the two tasks and categorize the gradient relationships into three categories: "conflict" (similarity between -1 and -0.01), "neutral" (-0.01 to 0.01), and "enhance" (0.01 to 1). The baseline model shows a

a). Task Gradient Conflict Analysis    b). Task-Expert Heatmap Base MoE    c). Task-Expert Heatmap M⁴oE

Figure 6: (a). Gradients from different task heads may conflict, and our method has mitigated this issue. (b). Compared to basic Soft MoE, our method has a sparser expert-task dependency that may have helped alleviate the gradient conflict problem.

high proportion of conflicts, indicating significant gradient conflicts in the multi-modal, multi-task setting. In contrast, our M⁴oE model exhibits a much lower proportion of conflicts, demonstrating that our method effectively mitigates gradient conflicts. This improvement may be attributed to sparse task-expert dependency, where experts tend to specialize in different tasks. As shown in Fig. 6, we conduct a reduced experiment (16 experts and 7 tasks) and generate heatmaps of the expert-task assignment scores from the combine matrix to visualize the expert-task dependencies in both the baseline MoE model and our M⁴oE. Our method shows higher sparsity, enabling dynamic separation of task-specific parameters and helping to reduce gradient conflicts.

## 4.4 ABLATION STUDY

**Ablation of Different Model Components.** As shown in Table 2, we conduct ablation studies on each component of the M⁴oE framework. Removing either the MSoE or MToE leads to a significant performance drop. This is because: 1) the MSoE retains and contributes modality-specific features to the embedding pool and alleviates the modality competition issue. 2) The task-slot and task-specific embedding guided MToE data flow capture the dependence between modalities and tasks to find the most suitable modality feature selection and fusion for each task. Additionally, removing the mutual-information regularization loss while retaining the MToE results in a slight performance decrease compared to original M⁴oE model, as the loss encourages higher modality-task dependence and more diverse modality feature fusion.

**Increasing the number of experts and regularization degree $\alpha$ within a certain range improves the performance.** As shown in Fig. 7, increasing the number of experts in the MToE from 16 to 128 improves the model's performance. Additionally, raising the value of $\alpha$, which controls the degree of mutual information regularization, from 0.01 to 0.2 generally enhances the model's performance, but too heavy of a weight makes the predictions slightly worse. Detailed results are in the Appendix.

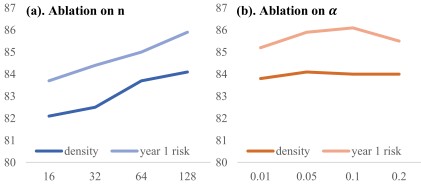

Figure 7: Ablation on expert number $n$ and regularization degree $\alpha$.

Table 2: Ablation results on technical components in M⁴oE.

| MToE | MSoE | MI Reg. | EMBED | | |
|---|---|---|---|---|---|
| | | | 1 year risk | Density | Birads |
| | ✓ | | 84.0 | 82.8 | 73.1 |
| ✓ | | ✓ | 85.1 | 83.5 | 73.8 |
| ✓ | ✓ | | 85.2 | 83.7 | 73.5 |
| ✓ | ✓ | ✓ | **85.9** | **84.1** | **75.1** |

## 5 CONCLUSION

In this paper, we propose M⁴oE, a Multi-modal Multi-task Mixture of Experts framework for Medical multi-modal multi-task learning. M⁴oE enables dynamic, sample-adaptive modality fusion and modality-task dependence modeling. It jointly models the dependence of modalities, experts, and tasks and captures dynamic distinct and shared modality information. Extensive experiments show that M⁴oE consistently outperforms baselines across diverse medical imaging benchmarks, offers interpretability of modality contribution, and can be flexibly combined with different backbones.

**Acknowledgements.** The authors acknowledge support from Michigan Institute for Computational Discovery and Engineering Catalyst Grant, and Michigan Institute for Data Science PODS Grant.

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
