In the supplementary materials, we discuss details of our analysis, datasets, and baselines. We also provide additional experiment results.

## A  DETAILS - ANALYSIS

### A.1  PRELIMINARY ON PARTIAL INFORMATION DECOMPOSITION THEORY

While classical information theory formalizes the amount of information that one variable provides about another, its direct extension to three or more variables remains a challenge (Zhang et al., 2024). Therefore, the Partial Information Decomposition (PID) theory is introduced to generalize the classical information theory to multiple variables (Liang et al., 2024). To be specific, let $\mathcal{X}_i, \mathcal{Y}$ be the sample spaces for features of the $i$-th modality and the labels of a certain task, and $\Delta$ be the set of joint distributions over $(\mathcal{X}_1, \mathcal{X}_2, \mathcal{Y})$. Given two features $X_1, X_2$ and labels $Y$ drawn from some distribution $p \in \Delta$, denote the total information that $X_1, X_2$ provide about task $Y$ as $I_p(X_1, X_2; Y)$, PID would decompose this total information into several parts, including:

- *Common Information*: The information shared between the two features $X_1, X_2$.
- *Modality Specific Information*: The information present in only $X_1$ or $X_2$, respectively.
- *Synergy Information*: The information that only emerges when both $X_1$ and $X_2$ are present.

Specifically, the formulation of synergy information is:

$$S = I_p(X_1, X_2; Y) - \min_{q \in \Delta_p} I_q(X_1, X_2, Y) \tag{1}$$

where $\Delta_p = \{q \in \Delta : q(x_i, y) = p(x_i, y) \; \forall y \in \mathcal{Y}, x_i \in \mathcal{X}_i, i \in \{1, 2\}\}$. In this work, we utilize the synergy information to measure the collaboration of different modalities on certain tasks, as shown in Figure 8.

### A.2  IMPLEMENTATION DETAILS OF PAIRWISE SYNERGY ESTIMATION VIA BATCH ALGORITHM

Pairwise synergy between modality $X_1$ and $X_2$, as defined in Equation 1, is the difference between total multi-modal information in $p$ and total multi-modal information in $q^* \in \Delta_p$, the non-synergistic distribution. Bertschinger et al. (2014); Liang et al. (2024; 2023) solves for $q$ distribution as a max-entropy optimization problem:

$$q^* = \arg\min_{q \in \Delta_p} I_q(\{X_1, X_2\}; Y) \tag{2}$$

We are tasked with finding the optimal distribution $q^*$ that minimizes the mutual information-like quantity $I_q$ over the joint variables $\{X_1, X_2\}$ and $Y$:

$$q^* = \arg\min_{q \in \Delta_p} I_q(\{X_1, X_2\}; Y)$$

This can be written as the following expectation:

$$q^* = \arg\min_{q \in \Delta_p} \mathbb{E}_{x_1, x_2, y \sim q} \left[ \log \frac{q(x_1, x_2, y)}{q(x_1, x_2) q(y)} \right]$$

Here, the mutual information is expressed as the expected logarithmic difference between the joint distribution $q(x_1, x_2, y)$ and the product of the marginals $q(x_1, x_2) q(y)$.

The conditional distribution $q(x_1, x_2 | y)$ is introduced next to simplify the joint distribution:

$$q^* = \arg\min_{q \in \Delta_p} \mathbb{E}_{x_1, x_2, y \sim q} \left[ \log \frac{q(x_1, x_2 | y)}{q(x_1, x_2)} \right]$$

Next, we factorize the conditional distribution $q(x_1, x_2 | y)$ into the product of conditional probabilities:

$$q^* = \arg\min_{q \in \Delta_p} \mathbb{E}_{x_1, x_2, y \sim q} \left[ \log \frac{q(x_2 | x_1, y) q(x_1 | y)}{q(x_2 | x_1) q(x_1)} \right]$$

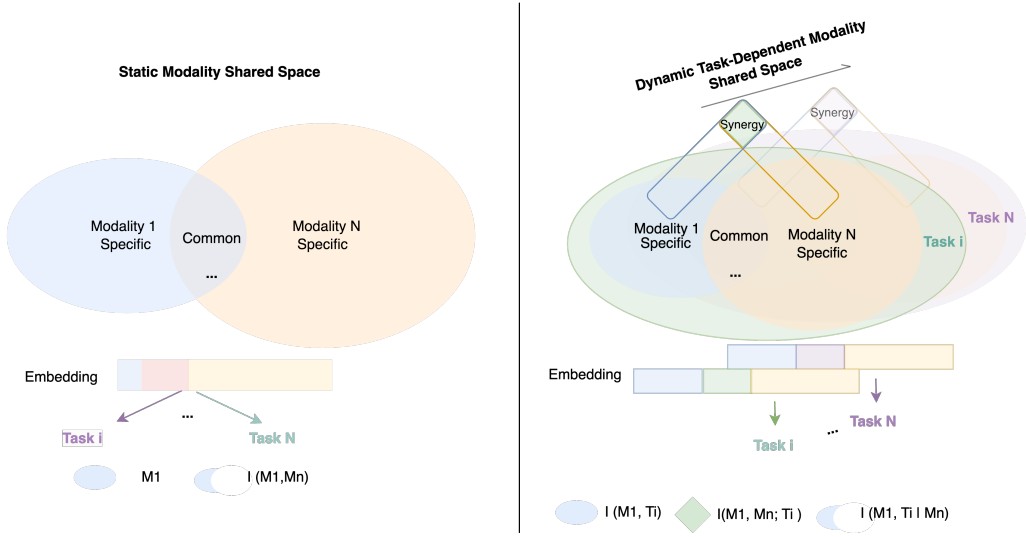

Figure 8: Illustration of dynamic modeling on modality-task dependence and traditional modeling of multi-modal fusion.

Finally, the expression can be marginalized over $x_1$ and conditioning on $y$:

$$q^* = \arg\min_{q \in \Delta_p} \mathbb{E}_{x_1, y \sim q(x_1, y), x_2 \sim q(x_2|x_1, y)} \left[ \log \frac{q(x_2|x_1, y)q(x_1|y)}{\sum_{y' \in Y} q(x_2|x_1, y')q(y'|x_1)q(x_1)} \right]$$

In this final form, the summation over all $y' \in Y$ appears in the denominator, representing the total probability distribution over $Y$. The goal is to minimize this log-probability difference, finding the optimal $q^*$.

In this equation, due to marginal constraints, $q(y'|x) = p(y'|x)$ which can be viewed as an unimodal model, $q(x_1, y)$ can be obtained by sampling from x1 and y labels in the real data distribution.

Given pairwise input modalities $\mathbf{X}_1 \in \tilde{\mathcal{X}}_1^z, \mathbf{X}_2 \in \tilde{\mathcal{X}}_2^z$, and task labels $\mathbf{Y} \in \mathcal{Y}^z$, where $z$ is batch size, we can represent the unnormalized joint distribution $\tilde{q}(x_1, x_2, y)$ by training two estimator $f$s that learns a outer-product similarity matrix over learned multi-modal features $A \in \mathbb{R}^{z \times z \times |\mathcal{Y}|}$. Here $A[i][j][y] = \tilde{q}(\mathbf{X}_1[i], \mathbf{X}_2[j], y)$ so that $A = \exp\left( f_{\phi(1)}(\mathbf{X}_1, y) f_{\phi(2)}(\mathbf{X}_2, y)^\top \right)$.

BATCH algorithm subsequently uses the Sinkhorn-Knopp (Distances, 2013) algorithm to constraint the learned $A$ to follow valid probability distributions. Sinkhorn-Knopp projects $A$ into the space of non-negative square matrices by iteratively normalizing all rows and columns of $A$ to sum to 1 and rescaling the rows and columns to satisfy the marginals. Sinkhorn's algorithm enables us to perform this projection By sampling $x_i$ from the dataset, the rows and columns of $A$ are already distributed as $p(x_i)$. The only remaining term needed is $p(y|x_i)$, for which we use unimodal models $\hat{p}(y|x_i)$ trained before running the estimator and subsequently frozen. Finally, each row is normalized to obtain $\hat{p}(y|x_1)$ and each column to $\hat{p}(y|x_2)$. Given a matrix A representing $\tilde{q}(x_1, x_2, y)$, we can obtain the remaining unknown terms in $q^*$ and optimize via gradient descent.

In practice, we need to first train separate unimodal models to obtain $\hat{p}(y \mid x_i)$, and a baseline model (here Mirai) and $M^4$oE as multi-modal models to obtain $\hat{p}(y \mid x_1, x_2)$. All these models are frozen and connected by two estimators, which we implement as shallow MLP layers, $f_1(.)$ and $f_2(.)$, to calculate the A matrix following the process described above.

Recall that synergy is $S = I_p(Y; X_1, X_2) - I_{\tilde{q}}(Y; X_1, X_2)$.

Now we have every term in

$$I_p(Y; X_1, X_2) = \mathbb{E}_{x_1, x_2, y \sim p} \left[ \log \left( \frac{\hat{p}(y \mid x_1, x_2)}{\hat{p}(y)} \right) \right]$$

and

$$I_{\tilde{q}}(Y; X_1, X_2) = \mathbb{E}_{x_1, x_2, y \sim \tilde{q}} \left[ \log \left( \frac{\tilde{q}(x_2 \mid x_1, y)\hat{p}(y \mid x_1)}{\hat{p}(y) \sum_{y'} \tilde{q}(x_2 \mid x_1, y')\hat{p}(y' \mid x_1)} \right) \right]$$

, we can then calculate the pairwise synergy of $x_1, x_2$ given $y$.

Based on our assumption of modality-task dynamic dependence, multi-modal models should exhibit different preferences when utilizing information from different modalities for different tasks. When combining two modalities that significantly contribute to a given task, the synergy between them should be higher as more information emerges from jointly leveraging these modalities. In multi-modal multi-task learning, models that primarily capture static modality-task correlations tend to rely on the same modalities for different tasks. As a result, the matrices of modality-pairwise synergies for different tasks are likely to be similar. For our M$^4$oE, which aims to capture dynamic modality-task dependence, the synergy matrices for different tasks tend to vary. This allows M$^4$oE to dynamically utilize information from different modalities during inference.

### A.3 IMPLEMENTATION DETAILS OF GRADIENT CONFLICT ANALYSIS

To investigate the gradient conflict problem, we gather gradients induced by different task losses during training and calculate the cosine similarity between them. Specifically, we trained a multi-modal, multi-task model in mammography using four modalities, focusing on two tasks. Each modality has a ViT-base (Dosovitskiy, 2020) as its own feature encoder. The input from each modality is fed into its corresponding encoder, and the resulting representations are gathered, concatenated, and then passed into the fusion block.

We collected the gradients from the last layer of the fusion block during back-propagation for both task losses in each step of an epoch. We calculate the cosine similarity between gradients of the two task losses for each step. In Fig. 6, we calculate the similarities of the gradient pairs of each step in epoch 5, and report the proportion of conflict pairs(similarity between (1,-0.01)), neural pairs(similarity between (-0.01,0.01)), and enhance pairs(similarity between (0.01,1)).

### A.4 IMPLEMENTATION DETAILS OF MODALITY COMPETITION ANALYSIS

To investigate our first hypothesis, we evaluate how traditional (non-MoE) multi-modal models, multi-modal MoE, and our approach utilize the four modalities on the EMBED dataset for the 1 year cancer risk prediction task. For traditional multi-modal learning, we first train a model where each modality is encoded by a modality-specific ViT-base (Dosovitskiy, 2020) encoder. The representations from each modality are averaged and passed through a fusion layer, implemented as a transformer block, before being fed into the task heads. We also trained four unimodal models for each modality. Each model consists of a Vit-base encoder and a classification head.

## B DETAILS - DATASETS

In this study, we have evaluated the methods on four public medical imaging benchmarks from mammography imaging and Ophthalmological screening. For mammography, we use the EMory BrEast Imaging Dataset (EMBED), the 2023 RSNA Screening Mammography Breast Cancer Detection AI Challenge Dataset (RSNA) (Carr et al.), and VinDr-Mammo Dataset (VinDr) (Nguyen et al., 2023). For Ophthalmological screening, we use the Glaucoma grAding from Multi-Modality imAges (GAMMA) dataset (Wu et al., 2023).

The EMory BrEast imaging Dataset (EMBED) is a large public screening dataset collected from two cohorts over an 8-year period containing 3.4M screening and diagnostic images from 110,000 patients. Four modalities used include: cranio-caudal (CC) and mediolateral-oblique (MLO) views of full-field digital mammography (FFDM), and CC and MLO views of synthesized 2D images from digital breast tomosynthesis (C-view). We cleaned and preprocessed the AWS release of the EMBED dataset which is a subset of the full dataset including 81775 records. Due to the fact that both VinDR and RSNA datasets have only three BI-RADS categories, to ensure a fair comparison with the BI-RADS tasks of VinDR and RSNA datasets, we reallocated the BI-RADS classifications

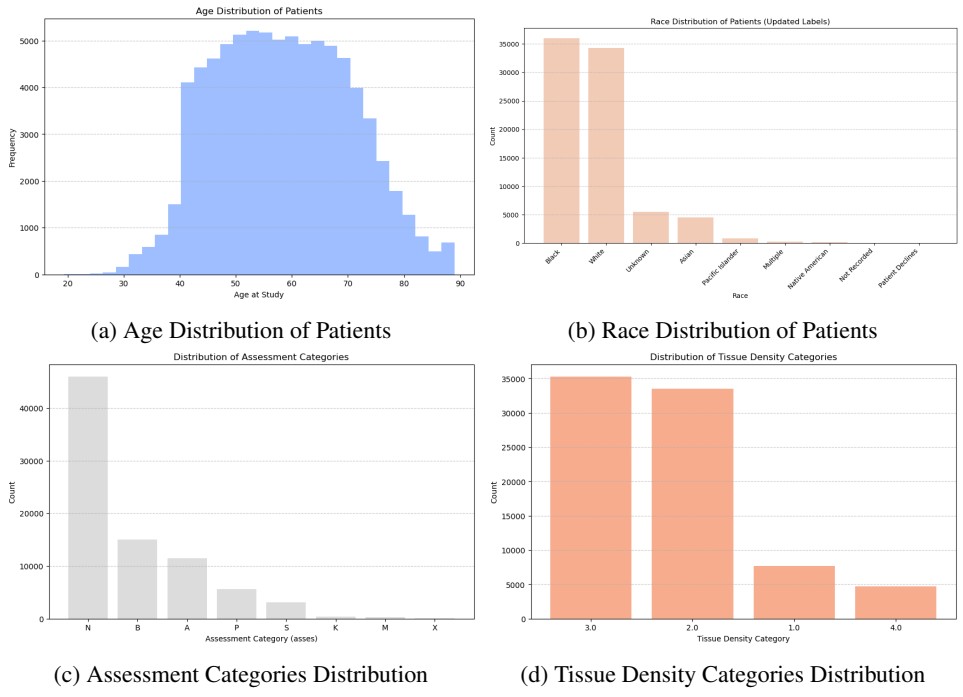

(a) Age Distribution of Patients  (b) Race Distribution of Patients

(c) Assessment Categories Distribution  (d) Tissue Density Categories Distribution

Figure 9: Visualization of Patient Data in EMBEDS dataset.

in the EMBED dataset into three categories following the method mentioned in the VinDR data. Specifically, BI-RADS 1, 2, and 3 were set as label A, BI-RADS 0 and 4 as label B, and BI-RADS 5 and 6 as label C. We visualize the dataset statistics below in Fig. 9.

The VinDr-Mammo dataset comprises 20,000 images from 5,000 patients, resulting in 10,000 visits. Each visit consists of CC and MLO views of FFDM for both breasts, accompanied by both a BI-RADS assessment and a density label. Detailed dataset statistics can be found in Nguyen et al. (2023).

The RSNA Screening Mammography Breast Cancer Detection dataset, originally from the AD-MANI dataset in Australia, includes 22,604 images from 5,651 patients with complete CC and MLO image modalities. Detailed dataset statistics can be found here on the AI challenge webpage.

The GAMMA Retinal dataset contains 500 paired colorful fundus images and OCT volumes from 450 patients, provided by Sun Yat-sen Ophthalmic Center, including both healthy subjects and those with various retinal diseases.

We employed a 5-fold cross-validation strategy with stratified sampling. Each fold was created by partitioning the dataset at the patient level into training and test sets with a ratio of 8:2, ensuring a balanced distribution of labels across all sets.

For all datasets, images were resized to 224x224 pixels during preprocessing. We carefully cleaned the data to ensure that images from the same patient did not appear across different splits within each fold. We employed stratified sampling to ensure that the distribution of key characteristics and labels remained consistent across the training and test sets in each fold.

Here, we will demonstrate the relationships between the characteristics of our benchmarks and challenges in Medical Multi-modal Multi-task Learning, which have been mentioned in Sec. 1.

**Challenge 1: Both modality-specific information and modality-shared information will be used for task prediction.** We have verified our method's capability of maintaining both modality-specific and shared information on our mammography and retinal benchmarks. In the mammography imaging domain, while all four imaging modalities can be used to classify breast density and predict the 5-Year risk, each modality provides specific information (Khara et al., 2024a). Specifically, FFDM excels in detecting microcalcifications and assessing breast density, while 2DS enhances the visibil-

ity of masses in dense tissue by reducing overlap (Aujero et al., 2017; Destounis et al., 2020), which makes it contribute more to the 5-Year risk prediction task.

**Challenge 2: Information contained in each modality varies differently across samples.** Our method creates data-dependent pathways for each input, enabling dynamic multi-modal fusion and effectively linking input modalities to specific tasks. We validate this on our medical imaging benchmarks, where the information in each modality varies across samples. In medical imaging, differences in devices, environments, and patient subgroups can significantly influence the information provided by each modality. For instance, in mammography screening, the information content of FFDM and 2DS images can vary significantly across patients due to differences in breast density. For patients with extremely dense breasts, FFDM may provide limited visibility of potential lesions due to tissue overlap  (Aujero et al., 2017; Zuckerman et al., 2016) , while 2DS images might offer better contrast and visibility of subtle architectural distortions or calcifications, whereas in patients with predominantly fatty breasts, FFDM might provide clearer anatomical details.

**Challenge 3: Different diagnostic tasks require a task-specific way to fuse features from different imaging modalities.** Our method jointly models the relationship of modalities, experts, and tasks, and has successfully addressed this challenge. We utilize the experiments on our medical imaging benchmarks to verify this point. Different modalities in medical imaging contribute differently across different tasks. For example, in a comprehensive breast cancer screening utilizing both FFDM and 2DS images, density assessment relies more heavily on FFDM due to its ability to provide clearer tissue details. Conversely, the cancer detection task, particularly for calcified lesions in dense tissue, benefits significantly from 2DS images due to their better visibility of such features, demonstrating how different diagnostic tasks require task-specific fusion of features from various imaging modalities.

## C  IMPLEMENTATION DETAILS

### C.1  $M^4$OE

Our method is implemented in PyTorch and runs on hardware consisting of 4 NVIDIA A100s and 4 L40s. In the main experiments, the model is trained using the Adam optimizer with an initial learning rate of 1e-4 and a StepLR scheduler (step size = 5, gamma = 0.1). We train the model with a batch size of 32 for a total of 100 epochs, and the loss hyperparameter $\alpha$ is set to 0.05. Early stopping is used to avoid over-fitting. We utilize 4 ViT-Base as modality-specific embedder as modality-specific encoders. [3] Each MoE block contains soft-MoE layer following Puigcerver et al. (2023), with 128 experts used for each MoE layer in the MToE module and 32 experts for each MoE layer in the MSoE modules. We utilize a MLP as the network of each expert.

For each task, before passing for final predictions, task-specific embeddings from MToE and task-shared embeddings from each MSoE together comprise a pool H. This H of shape [(1 (each task) + m)$\times$ 1 ]$\times$ d is again passed through a basic soft MoE block. Here this basic soft MoE block has 16 experts and 1 slot per expert, giving us a weighting matrix $\phi \in \mathbb{R}^{d\times 16\times 1}$. All task heads are MLP classifiers for classification tasks and we follow Chen et al. (2023c) for segmentation heads, please refer to github implementation for more details.

### C.2  MEDICAL AI SOTA

#### C.2.1  MAMMOGRAPHY

Mirai and AsymMirai are popular deep-learning frameworks for mammography-based short-term breast cancer risk prediction. Mirai consists of a convolutional neural network (CNN) and a transformer. It accepts the four standard screening mammography views, including the left and right mediolateral oblique and left and right craniocaudal views, as inputs to a ResNet-18 CNN backbone. After the features of each view are extracted by the CNN, they are further sent to the transformer part and finally give predictions of clinical risk factors and $n$-year breast cancer risk.

In contrast, AsymMirai excludes the transformer part in Mirai to maintain the spatial correspondence between the extracted features and input images. It instead computes a localized bilateral dissimi-

---

[3]Our code could be found in M4oE Official Implementation.

larity between the left and right breast at multiple locations, using those features for each view. The maximum dissimilarity accross locations would produce a dissimilarity score for each view, and the scores are averaged to produce one bilateral dissimilarity score. The outputs of AsymMirai could be directly overlayed on the mammogram to highlight dissimilarities.

For these two baselines, we use multiple task heads to replace the original prediction head to adapt them to multi-task learning scenes. We adapt our method to them by inserting the MToE block, and pass the fused representation into the MToE block(e.g., in Mirai, we feed the embedding projected by the transformer block into the MToE block).

### C.2.2 OPHTHALMOLOGY

We used EyeStar (Wu et al., 2023) and EyeMoSt (Zou et al., 2024a) as our ophthalmology baselines. Both methods employ a vision encoder to extract features from fundus photos and another vision encoder for OCT images. After feature extraction, EyeStar concatenates the features from each modality before passing them to the prediction head, while EyeMoSt uses a confidence-based teacher-student approach to fuse the features before feeding them into the task head.

We adapted these baselines for multi-task learning by replacing the single prediction head with multiple task heads. To upgrade these baselines with our method, we replaced the concatenation operation in EyeStar with our MToE block. In EyeMoSt, we inserted the MToE block before the confidence fusion stage, generating a set of task-specific representations. These representations were then passed through the confidence fusion stage and subsequently fed into the respective task heads.

### C.3 NATURAL DOMAIN MULTI-MODAL MULTI-TASK SOTA ADAPTED TO MEDICAL IMAGING

### C.3.1 EVIF

The EVIF model employs a multi-task collaborative framework to enhance the fusion quality of visible and infrared images using a transformer block and a bi-level min-max mutual information approach, resulting in a final fused image. Since their backbone network structure is quite different from ours, we adapt its mutual information loss and the transformer fusion block to our setting. Furthermore, we changed its final prediction head to multiple task heads for our multi-task learning setting.

### C.3.2 FULLER

Fuller is a multi-level gradient calibration learning framework designed to mitigate modality bias and task conflicts during optimization. It processes multi-modal inputs to perform both image recognition and segmentation tasks, utilizing shared backbone feature extractors for each modality. The extracted features are fused through a modality-fusion block and then passed to task-specific heads. To further address task conflicts and modality bias, Fuller applies multi-level gradient calibration throughout the optimization process. However, due to larger domain shifts in medical datasets, the shared backbone design may be less effective, resulting in slightly lower performance compared to natural domain baselines. For adaptation in our setting, we replace the original task heads with our custom task heads.

### C.3.3 AIDE

AIDE employs a feature-level multi-modal, multi-task fusion strategy to learn shared representations across multiple features and tasks, leveraging a cross-attention fusion module. This module uses a cross-attention mechanism to facilitate information interaction, enhancing each target feature effectively. For adaptation, we substitute the final prediction heads with our task-specific heads.

### C.3.4 MMODN

MModN is a multi-modal multi-task learning model that fuse latent representations sequentially, and providing real-time predictive feedback on multiple predictive tasks. The MModN architecture consists of state vectors, modality-specific encoders, and task-specific decoders. It starts with

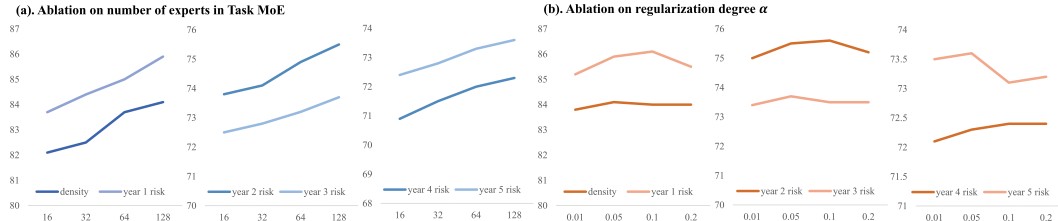

Figure 10: Ablation results on expert number $n$ and mutual information regularization degree $\alpha$.

an initial state $s_0$, which is updated sequentially by modality-specific encoders $e_i$, each taking in the previous state $s_{i-1}$ and producing an updated state $s_i$. These states can then be decoded by task-specific decoders $d_j$ to make predictions. Encoders represent individual modalities and can be skipped if a modality is missing. The model is agnostic to encoder and decoder types, supporting various architectures like Dense layers or CNNs. Decoders are used for different tasks, and their outputs are combined by averaging the loss. For adaptation to our setting, we replace the original task decoders to our multiple task heads.

### C.3.5 HYPERPARAMETERS

Ours: We determined optimal $\alpha$ weight for conditional mutual information loss and the number of experts $n$ based on the validation set conducted on the EMBED dataset. Then we keep using the same hyperparameters for our methods on all the benchmarks.

For baselines, we followed their experimental sections and ablation analysis to set the best-performing hyperparameters. We also keep the learning rate =1e-4 and batch size =32 fixed for all the baseline methods, including our approach, for a fair comparison. The hyperparameters used for baseline methods are: MultiModN: We set dropout rate = 0.1 and state representation size = 20.

EVIF: We set the weight $\gamma_1 = 1, \gamma_2 = 0.1, \gamma_3 = 0.01$ for the task loss, mutual information minimization loss, and mutual information maximization loss respectively. AIDE[6]: Instead of the original 1e-3 learning rate, we changed it to 1e-4 to keep comparisons fair.

Fuller: We set the weight factor $\alpha$ between modalities in gradient calibration to be 0.1. The gradient momentum hyperparameter m is set to 0.2.

AdaMVMoE: We set the auxiliary loss weights to be 5e-3.

## D LIMITATIONS

Our current work prioritizes establishing the foundational framework for dynamic modality-task modeling, with several promising directions for future extension. While our approach inherently supports training with missing labels due to its flexible task supervision scheme and could theoretically accommodate missing modalities through integration with embedding bank techniques (Yun et al., 2024), systematic investigation of these scenarios lies beyond our current scope.

## E ADDITIONAL EXPERIMENT RESULTS

Table 3: Ablation results on technical components in our proposed M$^4$oE.

| TMoE | SMoE | MI Reg. | EMBED | | | | | | |
|---|---|---|---|---|---|---|---|---|---|
| | | | 1 year risk | 2 year risk | 3 year risk | 4 year risk | 5 year risk | density | birads |
| | ✓ | | 84.0 | 73.8 | 72.1 | 70.9 | 71.5 | 82.8 | 73.1 |
| ✓ | | ✓ | 85.1 | 74.9 | 73.4 | 72 | 72.8 | 83.5 | 73.8 |
| ✓ | ✓ | | 85.2 | 75.2 | 73.3 | 72.4 | 73.0 | 83.7 | 73.5 |
| ✓ | ✓ | ✓ | 85.9 | 75.5 | 73.7 | 72.7 | 73.7 | 84.1 | 75.1 |

Table 4: Full experiment results on EMBED dataset.

| Setting | Method | EMBED | | | | | | |
|---|---|---|---|---|---|---|---|---|
| | | 1 year risk | 2 year risk | 3 year risk | 4 year risk | 5 year risk | density | birads |
| Single-Task | Mirai | 84.0 | 74.0 | 72.0 | 72.0 | 71.0 | 82.3 | 72.5 |
| | Asymirai | 79.0 | 69.0 | 69.0 | 67.0 | 66.0 | 80.2 | 69.4 |
| | wo MSoE | 83.2 | 74.5 | 73.0 | 71.8 | 71.2 | 82.1 | 71.9 |
| | M$^4$oE | 85.5 | 74.9 | 73.4 | 72.0 | 71.8 | 83.6 | 74.2 |
| Multi-Task | Mirai | 83.1 | 73.6 | 72.8 | 71.7 | 72.3 | 83.2 | 72.3 |
| | +MToE | 84.6 | 74.8 | 73.2 | 71.9 | 72.9 | 83.3 | 73.4 |
| | Asymirai | 80.0 | 68.6 | 67.9 | 68.4 | 66.5 | 82.5 | 69.6 |
| | +MToE | 81.9 | 70.2 | 69.0 | 69.9 | 68.4 | 83.1 | 70.8 |
| | EVIF | 84.7 | 75.7 | 72.4 | 70.9 | 72.4 | 83.4 | 73.6 |
| | Fuller | 84.3 | 74.4 | 71.9 | 70.0 | 71.5 | 83.5 | 72.5 |
| | AIDE | 84.1 | 74.7 | 72.5 | 71.7 | 72.5 | 82.9 | 73.6 |
| | MModN | 84.8 | 75.0 | 73.4 | 71.8 | 72.9 | 83.7 | 73.9 |
| | Wo MToE | 84.0 | 74.1 | 73.2 | 72.1 | 72.0 | 82.8 | 73.1 |
| | M$^4$oE | 85.9 | 75.5 | 73.7 | 72.7 | 73.7 | 84.1 | 75.1 |

## F  ADDITIONAL ANALYSIS RESULTS

Table 5: Analysis experiments on number of expert and mutual information regularization hyper-parameter $\alpha$.

| Num. of Expert $n$ | Reg. Degree $\alpha$ | EMBED | | | | | |
|---|---|---|---|---|---|---|---|
| | | 1 year risk | 2 year risk | 3 year risk | 4 year risk | 5 year risk | density |
| 16 | 0.05 | 83.7 | 73.8 | 72.5 | 70.9 | 72.4 | 82.1 |
| 32 | 0.05 | 84.4 | 74.1 | 72.8 | 71.5 | 72.8 | 82.5 |
| 64 | 0.05 | 85.0 | 74.9 | 73.2 | 72.0 | 73.3 | 83.7 |
| 128 | 0.01 | 85.2 | 75.0 | 73.4 | 72.1 | 73.5 | 83.8 |
| 128 | 0.05 | 85.9 | 75.5 | 73.7 | 72.7 | 73.7 | 84.1 |
| 128 | 0.10 | 86.1 | 75.6 | 73.5 | 72.4 | 73.1 | 84.0 |
| 128 | 0.20 | 85.5 | 75.2 | 73.5 | 72.4 | 73.2 | 84.0 |

Table 6: Pair-wise synergy matrix on density prediction task and risk prediction task.

| Modality | Density- Baseline | | | | Density - Ours | | | |
|---|---|---|---|---|---|---|---|---|
| | M1: FFDM U-D | M2: FFDM L-R | M3: Syn. U-D | M4: Syn. L-R | M1: FFDM U-D | M2: FFDM L-R | M3: Syn. U-D | M4: Syn. L-R |
| M1 | - | 0.030 | 0.016 | 0.008 | - | 0.041 | 0.012 | 0.004 |
| M2 | 0.030 | - | 0.021 | 0.017 | 0.041 | - | 0.013 | 0.003 |
| M3 | 0.016 | 0.021 | - | 0.006 | 0.012 | 0.013 | - | 0.005 |
| M4 | 0.008 | 0.017 | 0.006 | - | 0.004 | 0.003 | 0.005 | - |

| Modality | Risk - Baseline | | | | Risk - Ours | | | |
|---|---|---|---|---|---|---|---|---|
| | M1: FFDM U-D | M2: FFDM L-R | M3: Syn. U-D | M4: Syn. L-R | M1: FFDM U-D | M2: FFDM L-R | M3: Syn. U-D | M4: Syn. L-R |
| M1 | - | 0.071 | 0.057 | 0.030 | - | 0.076 | 0.059 | 0.071 |
| M2 | 0.071 | - | 0.032 | 0.041 | 0.076 | - | 0.048 | 0.044 |
| M3 | 0.057 | 0.032 | - | 0.045 | 0.059 | 0.048 | - | 0.055 |
| M4 | 0.030 | 0.041 | 0.045 | - | 0.071 | 0.044 | 0.055 | - |

## G  ADDITIONAL RESULTS

We have presented detailed results of the main table in Tables 7, Table 8, Table 9, and Table 11. We have reported the mean and standard deviation from five-fold cross-validation. The evaluation includes multiple metrics such as Accuracy, AUROC, and AUPRC, ensuring a comprehensive assessment.

Our method mainly focuses on modeling the dynamic multimodal fusion for multi-task learning, and is able to generalize to other modalities as well under this scheme. To demonstrate this, we performed additional multi-modal multi-task experiments on the MIMIC dataset following the experiment setting in Fuse-MoE. Specifically, we used X-ray images, clinical/radiology notes, and

electronic health records (EHR) modalities as inputs, to predict two tasks including in-hospital mortality prediction, and binary binned length of stay. The results are shown in Table 10. Through comparison with baseline methods, our method can achieve comparable results on mortality prediction and outperforms on length of stay prediction. Although this is not the main focus of our paper, we hope these additional experiments can convince that our proposed approach is potentially generalizable to non-imaging modalities as well.

We have also computed the modality-unique information following (Liang et al., 2024), as shown in the table below. The uniqueness measures how much unique information models derive from each modality given a certain dataset. We expect our model to alleviate the loss of unique information compared to baseline multimodal soft MoE methods. Compared to baseline MoEs, our M4oE achieves a higher uniqueness value for each modality, indicating that our approach effectively captures more modality-specific information. We add this metric in the revision in Table 12.

We report the computational cost of our model (inference time, training time, parameters) in Table 13. Note this analysis is conducted on a training batch size of 32 on the RSNA dataset. In the implementation, we used a single A100 80G GPU with 8 cores CPU to train the model. By comparing with other methods, our approach remains a similar time efficiency without adding latency, while using a similar scale of model parameters as other MoE methods. This is because the backbone of MoE introduces more model parameters than other structures (Puigcerver et al., 2023). MoE architectures in our method provide a promising way to scale the model size without paying too much computational cost. Our backbone, the recently introduced soft-MoE, runs at a faster speed than regular ViTs with 10x trainable parameters (Puigcerver et al., 2023). Similarly, M4oE can also scale to larger model sizes and datasets without introducing tremendous computational complexity.

We have provided ablation studies on RSNA, VinDR and GAMMA datasets in Table 14.

In order to investigate the sensitivity of our method on the number of experts, we also conduct an additional experiment: we removed all new task-related and modality-specific components proposed in our M4oE model to construct a basic version of multimodal MoE for comparison. The results are shown in Table 15. Compared to this basic MoE version, our method is less sensitive in the number of experts.

Whilst our experiment on EMBED yielded appearing even global modality contributions, the local modality contributions showed marked variation (Figure 5). In practice, we expect this pattern ought to differ across datasets and tasks. To further explore this behavior, we have conducted additional edge-case experiments to explicitly examine how our model manages severely compromised modalities. These experiments utilized the RSNA dataset with Gaussian noise corruptions of N(0, 1) * strength 4 and * strength 2 to simulate real-world scenarios where medical imaging modalities might be compromised due to poor acquisition or incorrect prescription. We maintained intact quality for both modalities in 70% of cases, whilst randomly corrupting one modality in the remaining 30%. Here we showcase example images showing the pre- and post-corruption states, demonstrating the compromised modality settings you suggested. We observe that the model is capable of utilizing the non-corrupted modalities. Interestingly, we also see that while the edge cases cause modality utilization to be highly biased towards the intact modalities, the overall contribution remains close due to data distribution. This could potentially help understand why the EMBED dataset shows even modality contributions.

Table 7: Detailed experiment results on the EMBED dataset.

| Setting | Method | EMBED, risk | | | EMBED, birads | | | EMBED, density | | |
|---|---|---|---|---|---|---|---|---|---|---|
| | | ACC | AUROC | AUPRC | ACC | AUROC | AUPRC | ACC | AUROC | AUPRC |
| Multimodal Single Task | Mirai | 0.840±0.033 | 0.769±0.039 | 0.653±0.050 | 0.725±0.021 | 0.701±0.014 | 0.598±0.019 | 0.823±0.030 | 0.898±0.027 | 0.751±0.034 |
| | Asymirai | 0.790±0.053 | 0.765±0.022 | 0.647±0.030 | 0.694±0.015 | 0.690±0.018 | 0.580±0.021 | 0.802±0.029 | 0.875±0.026 | 0.714±0.030 |
| | without MSoE | 0.832±0.026 | 0.768±0.014 | 0.650±0.043 | 0.719±0.030 | 0.695±0.024 | 0.585±0.037 | 0.821±0.019 | 0.898±0.013 | 0.752±0.026 |
| | M4oE (ours) | **0.855±0.041** | **0.791±0.030** | **0.663±0.034** | **0.742±0.027** | **0.705±0.033** | **0.603±0.028** | **0.836±0.034** | **0.900±0.039** | **0.768±0.025** |
| Multimodal Multi Task Medical Domain | Mirai | 0.831±0.047 | 0.776±0.054 | 0.662±0.043 | 0.723±0.021 | 0.708±0.019 | 0.605±0.027 | 0.832±0.026 | 0.894±0.028 | 0.763±0.021 |
| | Mirai+MToE | 0.846±0.031 | 0.781±0.050 | 0.671±0.070 | 0.734±0.028 | 0.712±0.031 | 0.607±0.023 | 0.833±0.030 | 0.897±0.022 | 0.771±0.029 |
| | Asymirai | 0.800±0.057 | 0.765±0.034 | 0.654±0.047 | 0.696±0.034 | 0.676±0.026 | 0.589±0.020 | 0.825±0.035 | 0.877±0.026 | 0.755±0.040 |
| | Asymirai+MToE | 0.819±0.061 | 0.771±0.059 | 0.665±0.056 | 0.708±0.019 | 0.682±0.030 | 0.602±0.029 | 0.831±0.027 | 0.889±0.014 | 0.760±0.021 |
| Multimodal Multi Task Natural Domain | EVIF | 0.847±0.034 | 0.813±0.038 | 0.682±0.024 | 0.736±0.022 | 0.719±0.035 | 0.619±0.031 | 0.834±0.019 | 0.906±0.017 | 0.773±0.023 |
| | Fuller | 0.843±0.021 | 0.793±0.035 | 0.681±0.045 | 0.725±0.057 | 0.718±0.047 | 0.622±0.053 | 0.835±0.033 | 0.909±0.034 | 0.778±0.038 |
| | AIDE | 0.841±0.018 | 0.790±0.026 | 0.678±0.026 | 0.736±0.032 | 0.721±0.028 | 0.623±0.035 | 0.829±0.022 | 0.891±0.020 | 0.771±0.030 |
| | MModN | 0.848±0.022 | 0.815±0.049 | 0.686±0.065 | 0.739±0.020 | 0.722±0.026 | 0.629±0.019 | 0.837±0.034 | 0.910±0.032 | 0.776±0.027 |
| | Multimodal Soft MoE | 0.833±0.012 | 0.779±0.014 | 0.665±0.015 | 0.721±0.031 | 0.702±0.024 | 0.593±0.038 | 0.815±0.030 | 0.872±0.024 | 0.750±0.036 |
| | AdaMV-MoE | 0.845±0.021 | 0.810±0.023 | 0.679±0.029 | 0.729±0.024 | 0.717±0.027 | 0.624±0.023 | 0.828±0.039 | 0.887±0.037 | 0.762±0.031 |
| | Fuse-MoE | 0.847±0.030 | 0.817±0.042 | 0.684±0.028 | 0.740±0.028 | 0.724±0.019 | 0.631±0.027 | 0.835±0.036 | 0.909±0.032 | 0.774±0.028 |
| | Our without MToE | 0.840±0.021 | 0.813±0.030 | 0.680±0.059 | 0.731±0.021 | 0.711±0.019 | 0.608±0.024 | 0.828±0.028 | 0.887±0.025 | 0.764±0.030 |
| | M4oE (ours) | **0.859±0.023** | **0.831±0.021** | **0.708±0.024** | **0.751±0.024** | **0.739±0.023** | **0.642±0.025** | **0.841±0.018** | **0.911±0.021** | **0.785±0.023** |

Table 8: Detailed experiment results on the RSNA dataset.

| Setting | Method | RSNA, density | | | RSNA, birads | | |
|---|---|---|---|---|---|---|---|
| | | ACC | AUROC | AUPRC | ACC | AUROC | AUPRC |
| Multimodal Single Task | Mirai | 0.763±0.029 | 0.824±0.026 | 0.635±0.039 | 0.623±0.020 | 0.682±0.022 | 0.553±0.020 |
| | Asymirai | 0.741±0.014 | 0.810±0.022 | 0.617±0.030 | 0.601±0.031 | 0.670±0.032 | 0.536±0.035 |
| | M4oE without MSOE | 0.768±0.048 | 0.832±0.042 | 0.642±0.040 | 0.622±0.024 | 0.680±0.019 | 0.550±0.023 |
| | M4oE (ours) | **0.775±0.022** | **0.838±0.016** | **0.644±0.035** | **0.640±0.017** | **0.700±0.013** | **0.576±0.017** |
| Multimodal Multi-task Medical AI | Mirai | 0.761±0.014 | 0.821±0.020 | 0.645±0.026 | 0.625±0.024 | 0.687±0.022 | 0.565±0.024 |
| | Mirai+MToE | 0.768±0.009 | 0.834±0.018 | 0.648±0.030 | 0.631±0.027 | 0.689±0.035 | 0.572±0.020 |
| | Asymirai | 0.739±0.020 | 0.805±0.016 | 0.638±0.024 | 0.607±0.024 | 0.674±0.030 | 0.555±0.027 |
| | Asymirai+MToE | 0.742±0.012 | 0.816±0.020 | 0.645±0.041 | 0.612±0.025 | 0.686±0.027 | 0.568±0.023 |
| Multimodal Multi-task Natural Domain | EVIF | 0.766±0.014 | 0.835±0.024 | 0.674±0.030 | 0.659±0.036 | 0.712±0.033 | 0.596±0.030 |
| | Fuller | 0.769±0.016 | 0.829±0.014 | 0.673±0.021 | 0.654±0.029 | 0.713±0.025 | 0.597±0.027 |
| | AIDE | 0.768±0.022 | 0.826±0.023 | 0.672±0.020 | 0.661±0.021 | 0.709±0.020 | 0.589±0.019 |
| | MModN | 0.771±0.018 | 0.834±0.020 | 0.675±0.025 | 0.664±0.023 | 0.718±0.026 | 0.595±0.026 |
| | Multimodal Soft MoE | 0.762±0.020 | 0.821±0.024 | 0.646±0.027 | 0.638±0.031 | 0.691±0.025 | 0.575±0.029 |
| | AdaMV-MoE | 0.767±0.018 | 0.824±0.022 | 0.670±0.030 | 0.655±0.019 | 0.706±0.021 | 0.584±0.020 |
| | Fuse-MoE | 0.771±0.024 | 0.834±0.026 | 0.678±0.037 | 0.663±0.027 | 0.714±0.032 | 0.592±0.035 |
| | Ours without MToE | 0.767±0.022 | 0.823±0.014 | 0.671±0.024 | 0.643±0.027 | 0.701±0.024 | 0.579±0.021 |
| | M4oE (ours) | **0.778±0.012** | **0.842±0.015** | **0.682±0.022** | **0.667±0.020** | **0.720±0.017** | **0.601±0.015** |

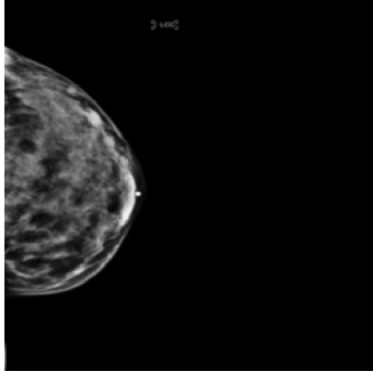
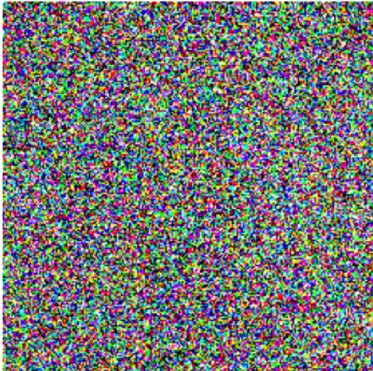

Figure 11: Comparison of images before and after corruption, Gaussian noise 4*N(0,1)

Table 9: Detailed experiment results on Vindr Dataset.

| Setting | Method | VinDR, Density | | | VinDR, Birads | | |
|---|---|---|---|---|---|---|---|
| | | ACC | AUROC | AUPRC | ACC | AUROC | AUPRC |
| Multimodal | Mirai | 0.863±0.025 | 0.824±0.014 | 0.717±0.011 | 0.661±0.023 | 0.705±0.014 | 0.571±0.010 |
| Single Task | Asymirai | 0.789±0.012 | 0.742±0.032 | 0.645±0.012 | 0.624±0.025 | 0.680±0.016 | 0.537±0.017 |
| | M4oE without MSOE | 0.854±0.015 | 0.814±0.014 | 0.661±0.016 | 0.658±0.017 | 0.703±0.012 | 0.569±0.016 |
| | M4oE (ours) | **0.877±0.019** | **0.857±0.012** | **0.753±0.020** | **0.675±0.019** | **0.714±0.015** | **0.590±0.017** |
| Multimodal | Mirai | 0.854±0.017 | 0.822±0.028 | 0.708±0.018 | 0.659±0.014 | 0.702±0.017 | 0.564±0.016 |
| Multi-task | Mirai+MToE | 0.859±0.014 | 0.831±0.013 | 0.714±0.012 | 0.664±0.013 | 0.710±0.016 | 0.579±0.012 |
| Medical AI | Asymirai | 0.791±0.015 | 0.752±0.022 | 0.654±0.011 | 0.622±0.025 | 0.671±0.012 | 0.528±0.018 |
| | Asymirai+MToE | 0.812±0.016 | 0.780±0.016 | 0.667±0.016 | 0.642±0.017 | 0.690±0.015 | 0.552±0.021 |
| Multimodal | EVIF | 0.888±0.025 | 0.869±0.021 | 0.760±0.019 | 0.707±0.021 | 0.734±0.026 | 0.620±0.019 |
| Multi-task | Fuller | 0.871±0.017 | 0.856±0.015 | 0.750±0.013 | 0.687±0.014 | 0.722±0.035 | 0.602±0.023 |
| Natural Domain | AIDE | 0.882±0.017 | 0.867±0.017 | 0.755±0.025 | 0.698±0.019 | 0.726±0.016 | 0.609±0.014 |
| | MModN | 0.890±0.018 | 0.878±0.021 | 0.765±0.023 | 0.704±0.017 | 0.732±0.015 | 0.613±0.018 |
| | Multimodal Soft MoE | 0.849±0.020 | 0.806±0.019 | 0.664±0.021 | 0.653±0.022 | 0.700±0.026 | 0.564±0.019 |
| | AdaMV-MoE | 0.880±0.016 | 0.861±0.015 | 0.754±0.018 | 0.679±0.018 | 0.712±0.016 | 0.595±0.014 |
| | Fuse-MoE | 0.892±0.028 | 0.881±0.020 | 0.764±0.017 | 0.705±0.026 | 0.733±0.018 | 0.618±0.019 |
| | Ours without MToE | 0.869±0.020 | 0.848±0.018 | 0.744±0.018 | 0.673±0.020 | 0.716±0.019 | 0.588±0.016 |
| | M4oE (ours) | **0.896±0.017** | **0.888±0.014** | **0.772±0.016** | **0.718±0.015** | **0.739±0.013** | **0.629±0.018** |

Table 10: Experiment results of multi-modal multi-task learning on the MIMIC dataset. We utilized EHR, Notes and CXR images modalities.

| Method | In-hospital Mortality AUROC | In-hospital Mortality AUPRC | Length of Stay AUROC | Length of Stay AUPRC |
|---|---|---|---|---|
| HAIM | 0.809±0.054 | 0.469±0.063 | 0.817±0.035 | 0.760±0.047 |
| EVIF | 0.818±0.046 | 0.519±0.055 | 0.816±0.037 | 0.762±0.041 |
| MModN | 0.814±0.031 | 0.522±0.039 | 0.824±0.032 | 0.778±0.028 |
| Fuse-MoE | 0.833±0.033 | 0.542±0.034 | 0.832±0.026 | 0.784±0.024 |
| M4oE (Ours) | **0.831±0.039** | **0.537±0.041** | **0.856±0.025** | **0.789±0.037** |

Table 11: Detailed results on the GAMMA dataset.

| Setting | Method | ACC | AUROC | AUPRC |
|---|---|---|---|---|
| Multimodal | Eyemost | 0.860±0.017 | 0.910±0.018 | 0.851±0.022 |
| Single Task | Eyestar | 0.854±0.029 | 0.906±0.022 | 0.841±0.032 |
| | M4oE without MSOE | 0.862±0.018 | 0.912±0.026 | 0.854±0.025 |
| | M4oE (ours) | **0.876±0.025** | **0.927±0.041** | **0.865±0.026** |
| Multimodal | Eyemost | 0.865±0.016 | 0.921±0.022 | 0.858±0.027 |
| Multi-task | Eyemost+MToE | 0.872±0.014 | 0.924±0.014 | 0.861±0.008 |
| Medical AI | Eyestar | 0.859±0.022 | 0.899±0.032 | 0.845±0.033 |
| | Eyestar+MToE | 0.875±0.013 | 0.926±0.008 | 0.865±0.011 |
| Multimodal | EVIF | 0.887±0.017 | 0.936±0.018 | 0.884±0.016 |
| Multi-task | Fuller | 0.878±0.021 | 0.927±0.019 | 0.877±0.022 |
| Natural Domain | AIDE | 0.885±0.029 | 0.933±0.025 | 0.881±0.030 |
| | MModN | 0.892±0.016 | 0.943±0.013 | 0.885±0.014 |
| | Multimodal Soft MoE | 0.871±0.029 | 0.918±0.027 | 0.863±0.025 |
| | AdaMV-MoE | 0.874±0.028 | 0.923±0.026 | 0.860±0.021 |
| | Fuse-MoE | 0.886±0.020 | 0.934±0.018 | 0.883±0.019 |
| | Ours without MToE | 0.881±0.022 | 0.930±0.039 | 0.879±0.026 |
| | M4oE (ours) | **0.904±0.017** | **0.952±0.015** | **0.895±0.018** |

Table 12: Comparison of the Uniqueness value of the baseline Multimodal Soft MoE and our M4oE methods on different modalities.

| Method | Uniqueness of M1 | Uniqueness of M2 | Uniqueness of M3 | Uniqueness of M4 |
|---|---|---|---|---|
| Multimodal Soft MoE | 0.124 | 0.111 | 0.068 | 0.072 |
| M4oE | 0.136 | 0.123 | 0.129 | 0.116 |

Table 13: Comparison of compuation complexity on inference time, training time, and parameter scales across different methods.

| Metric | Mirai | AsymMirai | EVIF | Fuller | AIDE | Ours | Fuse-MoE |
|---|---|---|---|---|---|---|---|
| Inference time/batch (s) | 0.37 | 0.38 | 0.50 | 0.35 | 0.36 | 0.47 | 0.44 |
| Training time/batch (s) | 1.15 | 1.01 | 1.56 | 0.98 | 1.12 | 1.51 | 1.32 |
| Parameters (M/B) | 120.4M | 205.4M | 188.1M | 211.6M | 254.3M | 3.2B | 2.1B |

Table 14: Additional ablation study results.

| Method | EMBED | | | RSNA | | VinDR | | GAMMA | |
|---|---|---|---|---|---|---|---|---|---|
| | Risk | Density | Birads | Density | Birads | Density | Birads | 3-class | Segmentation |
| No SMoE, No MI Reg | 0.840 | 0.828 | 0.731 | 0.767 | 0.643 | 0.869 | 0.673 | 0.881 | 0.874 |
| No MI Reg | 0.851 | 0.835 | 0.738 | 0.771 | 0.659 | 0.883 | 0.687 | 0.895 | 0.870 |
| No SMoE | 0.852 | 0.837 | 0.735 | 0.775 | 0.664 | 0.891 | 0.706 | 0.900 | 0.889 |
| Full Model | 0.859 | 0.841 | 0.751 | 0.778 | 0.667 | 0.896 | 0.718 | 0.904 | 0.897 |

Table 15: Sensitivity study on the number of experts.

| n | Method | Risk1 | Risk2 | Risk3 | Risk4 | Risk5 | Density |
|---|---|---|---|---|---|---|---|
| 16 | Ours | 83.7 | 73.8 | 72.5 | 70.9 | 72.4 | 82.1 |
| 32 | Ours | 84.4 | 74.1 | 72.8 | 71.5 | 72.8 | 82.5 |
| 64 | Ours | 85.0 | 74.9 | 73.2 | 72.0 | 73.3 | 83.7 |
| 128 | Ours | 85.9 | 75.5 | 73.7 | 72.7 | 73.7 | 84.1 |
| 16 | Baseline MoE | 80.9 | 71.7 | 71.2 | 69.5 | 70.6 | 81.3 |
| 32 | Baseline MoE | 82.1 | 72.5 | 71.7 | 70.4 | 71.4 | 82.0 |
| 64 | Baseline MoE | 83.0 | 73.2 | 72.3 | 71.2 | 72.0 | 82.7 |
| 128 | Baseline MoE | 83.8 | 73.7 | 72.9 | 72.0 | 72.4 | 83.3 |

Table 16: Edge Case (Modality Corruption Analysis) on RSNA Dataset

| **Corruption with Gaussian Noise 4 * N(0, 1)** | | | | |
|---|---|---|---|---|
| | Test Conditions | | | |
| | Overall Test | | Subgroup Test | |
| | All (100%) | 70% Uncorrupted Data | 15% Corrupted Modality A | 15% Corrupted Modality B |
| Density - Accuracy | 0.761 | 0.779 | 0.712 | 0.727 |
| BI-RADS - Accuracy | 0.645 | 0.666 | 0.608 | 0.584 |
| Density - Modality Contribution % | [0.537,0.463] | [0.539,0.461] | [0.195,0.805] | [0.868,0.132] |
| BI-RADS - Modality Contribution % | [0.425,0.575] | [0.408,0.592] | [0.157,0.843] | [0.773,0.227] |
| **100% Both Modality Corrupted** | | | | |
| Density - Accuracy | | 0.453 | | |
| BI-RADS - Accuracy | | 0.418 | | |
| **Corruption with Gaussian Noise 2 * N(0, 1)** | | | | |
| | Test Conditions | | | |
| | Overall Test | | Subgroup Test | |
| | All (100%) | 70% Uncorrupted Data | 15% Corrupted Modality A | 15% Corrupted Modality B |
| Density - Accuracy | 0.772 | 0.782 | 0.736 | 0.744 |
| BI-RADS - Accuracy | 0.652 | 0.667 | 0.615 | 0.603 |
| Density - Modality Contribution | [0.538,0.462] | [0.553,0.447] | [0.329,0.671] | [0.679,0.321] |
| BI-RADS - Modality Contribution | [0.489,0.511] | [0.485,0.515] | [0.348,0.652] | [0.646,0.354] |
| **100% Both Modality Corrupted** | | | | |
| Density - Accuracy | | 0.664 | | |
| BI-RADS - Accuracy | | 0.535 | | |