# OpenReview forum: "Dynamic Modeling of Patients, Modalities and Tasks via Multi-modal Multi-task Mixture of Experts"
_ICLR.cc/2025/Conference — ICLR 2025 Poster_

### Official Review · Reviewer_7L3d · 2024-11-02

**Soundness:** 3
**Presentation:** 3
**Contribution:** 2
**Rating:** 6
**Confidence:** 4

**Summary:**

This paper introduces M4oE, a framework for multi-modal, multi-task learning in medical diagnosis. M4oE addresses two primary challenges in multimodal diagnosis: sample-dynamic modality fusion and modality-task dependence. The framework incorporates modality-specific modules and a modality-shared modality-task mixture of experts (MoE) to dynamically learn both unique and shared information across modalities. A conditional mutual information loss is used to optimize the framework efficiently. Experimental results on two medical diagnostic tasks demonstrate M4oE’s advantages over existing methods.

**Strengths:**

1. The M4oE framework is a novel contribution, with two innovative components—modality-specific modules and a modality-shared modality-task MoE—that allow for efficient learning of both distinct and shared information across modalities for multiple tasks.

2. The analysis is comprehensive, including detailed evaluations of modality competition, modality-task dependence, and sample-level modality contributions.

**Weaknesses:**

1. The M4oE framework is complex, raising concerns about optimization and practical implementation. Including a discussion on computational resources and runtime would help readers assess its feasibility for real-world applications.

2. The paper does not address whether M4oE can function effectively when certain modalities or tasks are unavailable—a common scenario in clinical settings. Clarifying this would strengthen the model’s applicability.

**Questions:**

1. Could the authors provide computational resource requirements and runtime comparisons for M4oE and baseline methods?

2. In Figure 4, the similar difference between subfigures (a) and (b) and between (c) and (d) suggests both M4oE and the baseline might be capturing modality-task dependence. Could the authors clarify?

3. In Figure 3, is it reasonable to assume that the diverse distribution in (c) is preferable to that in (b), given the absence of a known ground truth distribution for each modality? Using the method in Liang et al. (2024) to quantify modality interactions could provide a more rigorous evaluation.

4. In Figure 1(a), under Part 3, should the label for the green triangle be $G_p$?

---

> ### Author Response · Authors · 2024-11-20
>
> We thank the reviewer for their valuable feedback. We updated our manuscript with some changes in writing and all additional results in Appendix Section F. Changes are marked in blue.
>
> **W1: “Computational complexity"**
>
> R1:  We report the computational cost of our model (inference time, training time, parameters) in Table 16. Note this analysis is conducted on a training batch size of 32 on the RSNA dataset. In the implementation, we used a single A100 80G GPU with 8 cores CPU to train the model. By comparing with other methods, our approach remains a similar time efficiency without adding latency, while using a similar scale of model parameters as other MoE methods. This is because the backbone of MoE introduces more model parameters than other structures [4].
>
> MoE architectures in our method provide a promising way to scale the model size without paying too much computational cost. Our backbone, the recently introduced soft-MoE, runs at a faster speed than regular ViTs with 10x trainable parameters[4]. Similarly, M4oE can also scale to larger model sizes and datasets without introducing tremendous computational complexity.
>
> Table 16 Complexity
>
> | Complexity           | mirai  | asymmirai | EVIF   | Fuller | AIDE   | Ours | Fuse-MoE |
> | -------------------- | ------ | --------- | ------ | ------ | ------ | ---- | -------- |
> | inference time/batch | 0.37   | 0.38      | 0.5    | 0.35   | 0.36   | 0.47 | 0.44     |
> | training time/batch  | 1.15   | 1.01      | 1.56   | 0.98   | 1.12   | 1.51 | 1.32     |
> | params               | 120.4M | 205.4M    | 188.1M | 211.6M | 254.3M | 3.2B | 2.1B     |
>
>
> **W2: “Missing modality”**
>
> R2:
> We agree that the missing modalities or missing labels could be interesting problems to solve. Although it is out of the scope for this paper, where we mainly focus on showing the proof-of-concept for dynamic modality-task modeling using the proposed method, our approach can flexibly deal with with the missing labels, since the model does not require the label supervision from all the tasks in each training iteration. Additionally, our method can also be extended with missing modalities by easily combining with missing modality techniques like learnable embedding banks [3]. These could both be interesting future works following this paper.
>
> We have added this discussion in the Discussion section 4.3.1 Of the revised manuscript.
>
>
> **Q1: “Computational resource requirements”**
>
> R3: We have reported the computational requirements in the revision. Please refer to our response R1 for details.
>
> **Q2: “Figure 4 clarification”**
>
> R4:  The synergies of baseline and our method in Figure 4 implies not only the difference in absolute values, but also different correlated patterns of high-synergy modality pairs, which indicates different modality-task dependency.
>
> Baseline: While Task 1 (density) and Task 2 (risk) synergy for baseline do capture different absolute values, the highest synergy pair still only peaks for M1 and M2 (within two views of FFDM) for both tasks.
>
> Ours: For density, our method peaks at M1-M2. For risk, we peak at not only M1-M2 synergy (within FFDM) but also M1-M4 synergy (between FFDM and 2DS). We also capture the M2-M3 and M3-M4 synergy much better than the baseline. This indicates a stronger modality-task dependency.
>
> Also, this result makes more sense from a clinical perspective. The FFDM (M1, M2) matters more for density predictions (Task 1) [7,8,9], while all four modalities indicate meaningful shared information for risk predictions (Task 2) [5, 6].
> We revised Sec. 4.3 to clarify this.
>
> **Q3: “Provide more PID theory experiments”**
>
> R5: Even though we do not know the ground truth distribution in this dataset, a combination of both FFDM and 2DS has been proven to be highly effective in different populations [10,11].
>
> As suggested by the reviewer, we have also computed the modality-unique information following [1], as shown in the table below. The uniqueness measures how much unique information models derive from each modality given a certain dataset. We expect our model to alleviate the loss of unique information compared to baseline multimodal soft MoE methods.
>
> Compared to baseline MoEs, our M4oE achieves a higher uniqueness value for each modality, indicating that our approach effectively captures more modality-specific information. We add this metric in the revision in Table 15.
>
> **Q4: “Figure typo”**
>
> R6: Thanks for catching this typo! We have corrected this in the manuscript.
>
> Table 15 Uniqueness Comparison
>
> | Uniqueness          | M1    | M2    | M3    | M4    |
> | ------------------- | ----- | ----- | ----- | ----- |
> | Multimodal Soft MoE | 0.124 | 0.111 | 0.068 | 0.072 |
> | M4oE                | 0.136 | 0.123 | 0.129 | 0.116 |

---

> ### Author Response · Authors · 2024-11-20
>
> [1] Liang P P, Cheng Y, Fan X, et al. Quantifying & modeling multimodal interactions: An information decomposition framework[J]. Advances in Neural Information Processing Systems, 2024, 36.
>
> [2] Soenksen L R, Ma Y, Zeng C, et al. Integrated multimodal artificial intelligence framework for healthcare applications[J]. NPJ digital medicine, 2022, 5(1): 149.
>
> [3] Yun, S., Choi, I., Peng, J., Wu, Y., Bao, J., Zhang, Q., ... & Chen, T. (2024). Flex-MoE: Modeling Arbitrary Modality Combination via the Flexible Mixture-of-Experts. arXiv preprint arXiv:2410.08245.
>
> [4]Puigcerver, J., Riquelme, C., Mustafa, B., & Houlsby, N. (2023). From sparse to soft mixtures of experts. arXiv preprint arXiv:2308.00951.
>
> [5] Nakajima, E., Tsunoda, H., Ookura, M., Ban, K., Kawaguchi, Y., Inagaki, M., ... & Ishikawa, T. (2021). Digital breast tomosynthesis complements two-dimensional synthetic mammography for secondary examination of breast cancer. Journal of the Belgian Society of Radiology, 105(1).
>
> [6] Mumin, N. A., Rahmat, K., Fadzli, F., Ramli, M. T., Westerhout, C. J., Ramli, N., ... & Ng, K. H. (2019). Diagnostic efficacy of synthesized 2D digital breast tomosynthesis in multi-ethnic Malaysian population. Scientific Reports, 9(1), 1459.
>
> [7] Brown, A. L., Vijapura, C., Patel, M., De La Cruz, A., & Wahab, R. (2023). Breast cancer in dense breasts: detection challenges and supplemental screening opportunities. RadioGraphics, 43(10), e230024.
>
> [8] Astley, S. M., Harkness, E. F., Sergeant, J. C., Warwick, J., Stavrinos, P., Warren, R., ... & Evans, D. G. (2018). A comparison of five methods of measuring mammographic density: a case-control study. Breast cancer research, 20, 1-13.
>
> [9] Heine, J. J., Fowler, E. E., & Flowers, C. I. (2011). Full field digital mammography and breast density: comparison of calibrated and noncalibrated measurements. Academic radiology, 18(11), 1430-1436.

---

> ### Author Response · Authors · 2024-11-25
>
> Dear Reviewer 7L3d,
>
> Thank you very much for your valuable feedback. We have provided comprehensive point-by-point responses to address each of your concerns and questions. As we approach the end of the discussion period, we would greatly appreciate your feedback on whether our responses have adequately addressed your points. We remain available to provide any additional clarification you may need.
>
> Best regards,
>
> Submission 11923 Authors

---

> ### Author Response · Authors · 2024-11-28
>
> Dear Reviewer 7L3d,
>
> Thanks very much for your time and valuable comments. We understand you're very busy. As the window for responding and paper revision is closing, would you mind checking our response and confirming whether you have any further questions? We are happy to provide answers and revisions to your additional questions. Many thanks!
>
> Best Regards,
>
> Authors of Submission 11923

---

> > ### Comment · Reviewer_7L3d · 2024-12-03
> >
> > Thank you for providing the additional information. I will maintain my current rating, and I appreciate your clarification.

---

> > > ### Author Response · Authors · 2024-12-03
> > >
> > > Dear Reviewer 7L3d,
> > >
> > > Thank you again for your thoughtful review and prompt response to our rebuttal. Please feel free to let us know if you have any remaining concerns. We welcome any additional feedback or questions you may have.
> > >
> > > Thank you for your time and consideration.
> > >
> > > Best Regards,
> > >
> > > Authors of Submission 11923

---

### Official Review · Reviewer_dm8E · 2024-11-03

**Soundness:** 2
**Presentation:** 3
**Contribution:** 2
**Rating:** 6
**Confidence:** 3

**Summary:**

This paper proposes a multi-modal, multi-task, mixture of experts for various medical diagnoses to address the challenges of sample-dynamic modality fusion (and modality-task dependence (selecting the right modalities for a task). Concretely, this is done by using a combination of modality-specific experts and experts shared between modalities and tasks. M4OE shows promising initial results in terms of both absolute performance and enforcing modality utilization.

Based on the weaknesses and questions outlined my score indicates a rejection for now, but I generally like the motivation of the paper, especially the aspect on modality utilization. I am willing to increase my score if my concerns are addressed and questions clarified.

**Strengths:**

- The M4OE is highly effective at enforcing modality utilization - this is a meaningful contribution that many multimodal models suffer from, although I do have some questions about this.
- The overall performance of the model is outperforming the baseline, even if the results are missing crucial information to validate the statistical significance of the results.
- Strong visuals that are additive to the understanding of the paper.
- Good conceptual motivation of the paper, although I believe that the motivation would further benefit from some concrete examples of sample dynamism and clinical examples of tasks that are modality-dependent.

**Weaknesses:**

- Abstract: the one-liner for sample-dynamic modality fusion is unclear as the specific and shared information always varies per sample unless they are identical. To my knowledge, sample-dynamic spans a much wider field of problems like missingness, robustness to noise, which the manuscript does not consider.
- Abstract: “Results demonstrate superiority over state-of-the-art methods” is extremely vague. Along which metric?
- You claim an expansive space by saying that the method is “multi-modal multi-task”, but your experiments only look at multi-view settings of a single modality (images). I would encourage you to narrow the scope/claim of the paper as the paper does not consider heterogeneous modalities (images, text, tabular, etc.).
- Experimental setup: I would encourage you to provide more detail in this section to aid reproducibility. For example, it is unclear whether cross-validation is used. No confidence intervals or standard deviation of results are reported to judge the statistical significance of the results. Additionally, no code was provided in the supplementary materials that would help with the clarification of the experimental setup.
- Literature: missing out on the largest corpus of literature (intermediate fusion), which many latent variable models for multimodal fusion fall under, many of which are using a mix of modality-specific and shared spaces.

**Questions:**

- How do you determine which expert sees which task? The connection between Figure 1 and the method section is not very clear.
- The manuscript talks a lot about sample adaptivity, but how does your experimental setup show that the model handles sample adaptivity effectively? Which aspects of sample adaptivity?
- Figure 3c suggests that the modality utilization is forced towards the same mean in your method. What about cases where modality dominance/competition is good? For example, if I have one very noisy modality, wouldn’t it be desirable to have the modality that contains all the signal to get all the model’s attention? Isn’t this graph showing that we enforce equal utilisation of all modalities regardless of the signal? Additionally, does this finding not contradict your claim in Figure 1, which is that only some experts are used (as opposed to all experts with a more balanced contribution).

---

> ### Author Response · Authors · 2024-11-20
>
> We thank the reviewer for their valuable feedback. We updated our manuscript with some changes in writing and all additional results in Appendix Section F. Changes are marked in blue.
>
> **W1: “About sample-dynamic modality fusion”**
>
> R1: Although we agree in practice there could be different dynamic cases for each patient sample including missing and noisy data, this work mainly focuses on discussing the dynamic modality-task dependency for different patient samples when modeling the multimodal fusion for multi-task prediction, instead of missing or noisy data. This is also why we name it as “sample-dynamic modality fusion”, because exactly as the reviewer already mentioned in the question: “the specific and shared information always varies per sample”.
>
> A similar term is also adopted in an analogous multimodal application in the natural domain for the dynamic fusion of infrared and RGB images [5,6]. For instance, in RGB-IR fusion, the relative importance of each modality naturally varies with environmental conditions - RGB provides better information in daylight while IR excels in low-light conditions. In mammogram screening, the optimal modality depends on patient characteristics: FFDM generally performs better for older patients while 2D synthesized images are more effective for younger patients [7,8]. So we think such a term is generally acceptable by the community in these applications.
>
> To make this definition more clear as well as the scope of the paper, we add these explanations to further clarify it in Sec. 1 Introduction, as highlighted in blue. We also added a limitation section on missing modality in Section 4.3.1.
>
>
> **W2: “Performance metrics”**
>
> R2: M4oE outperforms baselines with different performance metrics. We reported Accuracy in classification tasks and DICE score in segmentation tasks. We have now also included AUROC and AUPRC. We have revised this sentence in the updated manuscript to be more clear about this as: “ Results demonstrate superiority over state-of-the-art methods under different metrics of classification and segmentation tasks including Accuracy, AUROC, AUPRC, and DICE.” Please check updated results in Appendix F, table 9-13.
>
> **W3: “Claim on multimodality”**
>
> R3: In the mammography domain, we are using not only different views of images, but actually different imaging modalities acquired using totally different techniques.  Similarly, in the experiments on ophthalmology, the model adopts color fundus photos and optical coherence tomography, which are also two different imaging modalities acquired using different imaging systems. Unlike natural RGB images, medical image modalities can be much more complicated due to the essential difference in the imaging physics so as to capture distinct information in each modality. Therefore, we respectfully argue that the applications with these different imaging modalities should also essentially be multimodal problems. Previous works on the same applications also claim multimodal problems[11,12,13], which are already generally recognized by the community.
>
> Besides, to avoid confusion with other data modalities, we have also pointed out that this work was for “medical imaging”, when we introduced the problem in Line 47 and our method in Line 99.
>
> We agree that it’d be interesting to explore non-imaging modalities using our framework as well. Therefore, following your and Reviewer P7K2’s valuable advice, we performed additional multi-modal multi-task experiments on the MIMIC dataset following the experiment setting in [2][8]. Specifically, we used X-ray images, clinical/radiology notes, and electronic health records (EHR) modalities as inputs, to predict two tasks including in-hospital mortality prediction, and binary binned length of stay. The results are shown in Table 12.
>
> Through comparison with baseline methods, our method can achieve comparable results on mortality prediction and outperforms on length of stay prediction.

---

> > ### Author Response · Authors · 2024-11-20
> >
> > Table 11 VinDR Results
> >
> > | setting                  |                     | Vindr, density  |                   |                   | VinDR, birads   |                   |                   |
> > | ------------------------ | ------------------- | --------------- | ----------------- | ----------------- | --------------- | ----------------- | ----------------- |
> > |                          | Baselines           | ACC             | AUROC             | AUPRC             | ACC             | AUROC             | AUPRC             |
> > | Multimodal ，Single Task | Mirai               | 0.863±0.025     | 0.824 ± 0.014     | 0.717 ± 0.011     | 0.661±0.023     | 0.705 ± 0.014     | 0.571 ± 0.010     |
> > |                          | Asymirai            | 0.789±0.012     | 0.742 ± 0.032     | 0.645 ± 0.012     | 0.624±0.025     | 0.680 ± 0.016     | 0.537 ± 0.017     |
> > |                          | M4oE without MSOE   | 0.854±0.015     | 0.814 ± 0.014     | 0.661 ± 0.016     | 0.658±0.017     | 0.703 ± 0.012     | 0.569 ± 0.016     |
> > |                          | M4oE                | 0.877±0.019     | 0.857 ± 0.012     | 0.753 ± 0.020     | 0.675±0.019     | 0.714 ± 0.015     | 0.590 ± 0.017     |
> > | Multimodal Multi task    | Mirai               | 0.854±0.017     | 0.822 ± 0.028     | 0.708 ± 0.018     | 0.659±0.014     | 0.702 ± 0.017     | 0.564 ± 0.016     |
> > | (medical ai)             | Mirai+MToE          | 0.859±0.014     | 0.831 ± 0.013     | 0.714 ± 0.012     | 0.664±0.013     | 0.710 ± 0.016     | 0.579 ± 0.012     |
> > |                          | Asymirai            | 0.791±0.015     | 0.752 ± 0.022     | 0.654 ± 0.011     | 0.622±0.025     | 0.671 ± 0.012     | 0.528 ± 0.018     |
> > |                          | Asymirai+MToE       | 0.812±0.016     | 0.780 ± 0.016     | 0.667 ± 0.016     | 0.642±0.017     | 0.69 ± 0.015      | 0.552 ± 0.021     |
> > | Multimodal Multi-task    | EVIF                | 0.888±0.025     | 0.869 ± 0.021     | 0.760 ± 0.019     | 0.707±0.021     | 0.734 ± 0.026     | 0.62 ± 0.019      |
> > | (natural domain)         | Fuller              | 0.871±0.017     | 0.856 ± 0.015     | 0.75 ± 0.013      | 0.687±0.014     | 0.722 ± 0.035     | 0.602 ± 0.023     |
> > |                          | AIDE                | 0.882±0.017     | 0.867 ± 0.017     | 0.755 ± 0.025     | 0.698±0.019     | 0.726 ± 0.016     | 0.609 ± 0.014     |
> > |                          | MModN               | 0.890±0.018     | 0.878 ± 0.021     | 0.765 ± 0.023     | 0.704±0.017     | 0.732 ± 0.015     | 0.613 ± 0.018     |
> > |                          | Multimodal Soft Moe | 0.849 ± 0.02    | 0.806 ±  0.019    | 0.664 ±  0.021    | 0.653 ± 0.022   | 0.7 ± 0.026       | 0.564 ±  0.019    |
> > |                          | AdaMV-MoE           | 0.88 ± 0.016    | 0.861 ±  0.015    | 0.754 ±  0.018    | 0.679 ± 0.018   | 0.712 ±  0.016    | 0.595 ±  0.014    |
> > |                          | Fuse-MoE            | 0.892 ± 0.028   | 0.881 ±  0.02     | 0.764 ±  0.017    | 0.705 ± 0.026   | 0.733 ±  0.018    | 0.618 ±  0.019    |
> > |                          | Ours without MTOE   | 0.869±0.020     | 0.848 ± 0.018     | 0.744 ± 0.018     | 0.673±0.020     | 0.716 ± 0.019     | 0.588 ± 0.016     |
> > |                          | Ours                | **0.896±0.017** | **0.888 ± 0.014** | **0.772 ± 0.016** | **0.718±0.015** | **0.739 ± 0.013** | **0.629 ± 0.018** |

---

> ### Author Response · Authors · 2024-11-20
>
> **W4: “Experimental setup”**
>
> R4: Ours: We determined optimal $\alpha$ weight for conditional mutual information loss and the number of experts $n$ based on the validation set conducted on the EMBED dataset. Then we keep using the same hyperparameters for our methods on all the benchmarks.
>
> For baselines, we followed their experimental sections and ablation analysis to set the best-performing hyperparameters. We also keep the learning rate =1e-4 and batch size =32 fixed for all the baseline methods, including our approach, for a fair comparison. The hyperparameters used for baseline methods are:
>
> MModN[7] : We set dropout rate = 0.1 and state representation size = 20.
>
> EVIF[3]: We set the weight $\gamma_1=1,\gamma_2 =0.1, \gamma_3=0.01$ for the task loss, mutual information minimization loss, and mutual information maximization loss respectively.
>
> AIDE[6]: Instead of the original 1e-3 learning rate, we changed it to 1e-4 to keep comparisons fair.
>
> Fuller[5]: We set the weight factor $\alpha$ between modalities in gradient calibration to be 0.1. The gradient momentum hyperparameter m is set to 0.2.
>
> AdaMV-MoE[1]: We set the auxiliary loss weights to be 5e−3 and delta_n to be 2000.
>
> We have added content about hyperparameter tuning and settings in Appendix Sec. C.3.5  to further clarify this clearly. See changes in revision highlighted in blue.
>
> We hope all these additional implementation details can further help to enhance the reproducibility of our work. Besides, we will also share our code publicly upon the paper's acceptance.
>
> We have added the new metrics for our main experiments and reported AUROC and AUPRC in Appendix Tables 9-13. Along with the Accuracy, our proposed method achieves superior performance in different metrics.
>
> **W5: “Literature review misses intermediate fusion”**
>
> R5: Thanks for the constructive feedback! We agree that the papers about intermediate fusion can be relevant to our work in the perspective of multimodal fusion and feature disentanglement. But our work provides a unique insight for dynamic modality-task modeling parametrized by MoE structure and dynamic sample fusion. We also provide sample-level and population-level interpretability.
> We have added these discussions with relevant works brought up by Reviewer xATa in our related works section (Sec. 2) as highlighted in the updated revision.
>
>
>
> **Q1: “Expert-task assignment”**
>
> R6: Figure 1 provides a high-level schematic diagram showing only the abstraction of our key motivation: linking modality and tasks through experts. In our method, as shown in Fig. 2 and Sec.3, we have introduced that each expert contributes to every task but with different weight assignments determined by the combined matrix. Thus, the weight assignments are also learned automatically, enabled by the backbone of soft MoE [3].
>
> **Q2:“Sample adaptivity”**
>
> R7: Our work addresses sample adaptivity in dynamic multimodal medical imaging fusion, similar to well-established dynamic fusion scenarios in computer vision[6]. For instance, in RGB-Infared fusion, the relative importance of each modality naturally varies with environmental conditions - RGB provides better information in daylight while Infrared excels in low-light conditions. Analogously, in mammogram screening, the optimal modality depends on patient characteristics: FFDM generally performs better for older patients [7], while 2D synthesized images are more effective for younger patients [8].
>
> Effective sample adaptivity requires the model to selectively utilize the most relevant features from each modality based on the input. However, naive fusion approaches often result in one modality dominating the joint representation, causing the loss of valuable complementary information from other modalities. Our proposed method addresses this challenge by preserving distinct features from each modality while allowing dynamic, sample-specific fusion. Specifically, the soft MoE backbones for both MSoE and MToE are designed to create input-conditioned data pathways dependent on each input respectively[5]. The MSoE modules preserve modality-specific information, which prevents one modality from being underestimated.
>
> To qualitatively demonstrate the sample adaptivity, we discussed in original paper’s Fig 5 and Sec. 4.3 to show different modality contributions across samples.
>
>  We also provide the detailed performance of different subgroups, as shown below and in Table 14.

---

> ### Author Response · Authors · 2024-11-20
>
> **Q3: Interpretation of modality utilisation**
>
> R8:
> **How to interpret global vs local modality contribution**: Thanks for the reviewer’s questions. Let’s look at both Fig. 3(c) and Fig.5. Figure 3(c) demonstrates the global distribution of modality importance scores, while Fig. 5 shows both global (population-level) and local (sample-level) modality contributions. From Fig. 3(c), we see that M4oE does not heavily rely on one single modality globally, but captures dynamic dependency across all input modalities. This also aligns well with the average population-level results in Fig. 5, which shows that the modality contribution is close in average, with FFDM leads a bit. However, this does not mean the modality contribution is also purely equivalent for each sample. As shown in the sample-level results in Fig. 5, we can see the modality contribution is quite different for different individual patients. We also provide a clinical interpretation for these two patient cases in Sec. 4.3. This indicates that our model allows for dynamic diverging modality contribution for each patient sample in local modality contribution, while avoiding modality competition statistically in the sense of global modality contribution.
>
> **More on modality domination** : Different modalities should always carry out some useful information, but due to the limitations of the multimodal fusion modeling, the model may become over-reliant on one modality, making a statistical bias, as shown in the Fig. 3(b) for the case of plain Soft MoE. Our method can address this issue to leverage the useful information from different modalities more effectively. However, in the extreme case as the reviewer mentioned where there exists information bias among different modalities due to data quality, we conjecture the distribution across different modalities may be more different statistically in mean and variance. But this is absolutely not the case in the dataset used here, where we can still see the difference of mean values of different modalities but not diverge a lot (it is not forced to be the same in the model design). This actually aligns with the clinical interpretation in this dataset and application where a combination of both FFDM and 2DS is proven to be highly effective on different populations [10,11].
>
> Figure 1 provides a high-level schematic diagram showing only the abstraction of our key motivation: linking modality and tasks through experts. Each expert contributes to every task but with different weight assignments determined by the combined matrix. Also, in the right half of Figure 1, all experts are linked to all tasks.

---

> ### Author Response · Authors · 2024-11-20
>
> [1] Cao B, Sun Y, Zhu P, et al. Multi-modal gated mixture of local-to-global experts for dynamic image fusion[C]//Proceedings of the IEEE/CVF International Conference on Computer Vision. 2023: 23555-23564.
>
> [2] Yun, S., Choi, I., Peng, J., Wu, Y., Bao, J., Zhang, Q., ... & Chen, T. (2024). Flex-MoE: Modeling Arbitrary Modality Combination via the Flexible Mixture-of-Experts. arXiv preprint arXiv:2410.08245.
>
> [3] Puigcerver, J., Riquelme, C., Mustafa, B., & Houlsby, N. (2023). From sparse to soft mixtures of experts. arXiv preprint arXiv:2308.00951.
>
> [4] Han, X., Nguyen, H., Harris, C., Ho, N., & Saria, S. (2024). Fusemoe: Mixture-of-experts transformers for fleximodal fusion. arXiv preprint arXiv:2402.03226.
>
> [5] Cao, B., Sun, Y., Zhu, P., & Hu, Q. (2023). Multi-modal gated mixture of local-to-global experts for dynamic image fusion. In Proceedings of the IEEE/CVF International Conference on Computer Vision (pp. 23555-23564).
>
> [6] Zhang, Q., Wei, Y., Han, Z., Fu, H., Peng, X., Deng, C., ... & Zhang, C. (2024). Multimodal fusion on low-quality data: A comprehensive survey. arXiv preprint arXiv:2404.18947.
>
> [7] Destounis, S. V., Santacroce, A., & Arieno, A. (2020). Update on breast density, risk estimation, and supplemental screening. American Journal of Roentgenology, 214(2), 296-305.
>
> [8] McDonald, E. S., McCarthy, A. M., Akhtar, A. L., Synnestvedt, M. B., Schnall, M., & Conant, E. F. (2015). Baseline screening mammography: performance of full-field digital mammography versus digital breast tomosynthesis. American Journal of Roentgenology, 205(5), 1143-1148.
>
> [9] Mumin, N. A., Rahmat, K., Fadzli, F., Ramli, M. T., Westerhout, C. J., Ramli, N., ... & Ng, K. H. (2019). Diagnostic efficacy of synthesized 2D digital breast tomosynthesis in multi-ethnic Malaysian population. Scientific Reports, 9(1), 1459.
>
> [10] Nakajima, E., Tsunoda, H., Ookura, M., Ban, K., Kawaguchi, Y., Inagaki, M., ... & Ishikawa, T. (2021). Digital breast tomosynthesis complements two-dimensional synthetic mammography for secondary examination of breast cancer. Journal of the Belgian Society of Radiology, 105(1).
>
> [11] Khara, Galvin, et al. "Generalisable deep learning method for mammographic density prediction across imaging techniques and self-reported race." Communications Medicine 4.1 (2024): 21.
>
> [12] Cai, Zhiyuan, et al. "Uni4eye++: A general masked image modeling multi-modal pre-training framework for ophthalmic image classification and segmentation." IEEE Transactions on Medical Imaging (2024).
>
> [13] He, Xingxin, et al. "Multi-modal retinal image classification with modality-specific attention network." IEEE transactions on medical imaging 40.6 (2021): 1591-1602.

---

> ### Author Response · Authors · 2024-11-20
> **Table 9 Embeds results**
>
> Table 9 Embeds Results
>
> | Setting                  | Method              | EMBED, risk     |                   |                   | EMBED, birads   |                   |                   | EMBED, density   |                   |                   |
> | ------------------------ | ------------------- | --------------- | ----------------- | ----------------- | --------------- | ----------------- | ----------------- | ---------------- | ----------------- | ----------------- |
> |                          |                     | ACC             | AUROC             | AUPRC             | ACC             | AUROC             | AUPRC             | ACC              | AUROC             | AUPRC             |
> | Multimodal ，Single Task | Mirai               | 0.840±0.033     | 0.769 ± 0.039     | 0.653 ± 0.050     | 0.725±0.021     | 0.701 ± 0.014     | 0.598 ± 0.019     | 0.823± 0.030     | 0.898 ± 0.027     | 0.751 ± 0.034     |
> |                          | Asymirai            | 0.790±0.053     | 0.765 ± 0.022     | 0.647 ± 0.030     | 0.694±0.015     | 0.690 ± 0.018     | 0.580 ± 0.021     | 0.802± 0.029     | 0.875 ± 0.026     | 0.714 ± 0.03      |
> |                          | M4oE without MSOE   | 0.832±0.026     | 0.768 ± 0.014     | 0.650 ± 0.043     | 0.719±0.030     | 0.695 ± 0.041     | 0.585 ± 0.037     | 0.821± 0.019     | 0.898 ± 0.013     | 0.752 ± 0.026     |
> |                          | M4oE                | 0.855±0.041     | 0.791 ± 0.030     | 0.663 ± 0.034     | 0.742±0.027     | 0.705 ± 0.033     | 0.603 ± 0.028     | 0.836± 0.034     | 0.900 ± 0.039     | 0.768 ± 0.025     |
> | Multimodal Multi task    | Mirai               | 0.831±0.047     | 0.776 ± 0.054     | 0.662 ± 0.043     | 0.723±0.021     | 0.708 ± 0.019     | 0.605 ± 0.027     | 0.832± 0.026     | 0.894 ± 0.028     | 0.763 ± 0.021     |
> | (medical ai)             | Mirai+MToE          | 0.846±0.031     | 0.781 ± 0.050     | 0.671 ± 0.070     | 0.734±0.028     | 0.712 ± 0.031     | 0.607 ± 0.023     | 0.833± 0.030     | 0.897 ± 0.022     | 0.771 ± 0.029     |
> |                          | Asymirai            | 0.800±0.057     | 0.765 ± 0.034     | 0.654 ± 0.047     | 0.696±0.034     | 0.676 ± 0.026     | 0.589 ± 0.020     | 0.825± 0.035     | 0.877 ± 0.026     | 0.755 ± 0.040     |
> |                          | Asymirai+MToE       | 0.819±0.061     | 0.771 ± 0.059     | 0.665 ± 0.056     | 0.708±0.019     | 0.682 ± 0.030     | 0.602 ± 0.029     | 0.831± 0.027     | 0.889 ± 0.014     | 0.760 ± 0.021     |
> | Multimodal Multi-task    | EVIF                | 0.847±0.034     | 0.813 ± 0.038     | 0.682 ± 0.024     | 0.736±0.022     | 0.719± 0.035      | 0.619 ± 0.031     | 0.834± 0.019     | 0.906 ± 0.017     | 0.773 ± 0.023     |
> | (natural domain)         | Fuller              | 0.843±0.021     | 0.793 ± 0.035     | 0.681 ± 0.045     | 0.725±0.057     | 0.718 ± 0.047     | 0.622 ± 0.053     | 0.835± 0.033     | 0.909 ± 0.034     | 0.778 ± 0.038     |
> |                          | AIDE                | 0.841±0.018     | 0.790 ± 0.026     | 0.678 ± 0.026     | 0.736±0.032     | 0.721± 0.028      | 0.623± 0.035      | 0.829± 0.022     | 0.891 ± 0.020     | 0.771 ± 0.030     |
> |                          | MModN               | 0.848±0.022     | 0.815 ± 0.049     | 0.686 ± 0.065     | 0.739±0.020     | 0.722 ± 0.026     | 0.629 ± 0.019     | 0.837± 0.034     | 0.910 ± 0.032     | 0.776 ± 0.027     |
> |                          | Multimodal Soft Moe | 0.833±0.012     | 0.779±0.014       | 0.665±0.015       | 0.721±0.031     | 0.702±0.024       | 0.593±0.038       | 0.815±0.030      | 0.872±0.024       | 0.750±0.036       |
> |                          | AdaMV-MoE           | 0.845±0.021     | 0.810±0.023       | 0.679±0.029       | 0.729±0.024     | 0.717±0.027       | 0.624±0.023       | 0.828±0.039      | 0.887±0.037       | 0.762±0.031       |
> |                          | Fuse-MoE            | 0.847±0.030     | 0.817±0.042       | 0.684±0.028       | 0.740±0.028     | 0.724±0.019       | 0.631±0.027       | 0.835±0.036      | 0.909±0.032       | 0.774±0.028       |
> |                          | Ours without MTOE   | 0.840±0.021     | 0.813 ± 0.030     | 0.680 ± 0.059     | 0.731±0.021     | 0.711 ± 0.019     | 0.608 ± 0.024     | 0.828± 0.028     | 0.887± 0.025      | 0.764 ± 0.030     |
> |                          | Ours                | **0.859±0.023** | **0.831 ± 0.021** | **0.708 ± 0.024** | **0.751±0.024** | **0.739 ± 0.023** | **0.642 ± 0.025** | **0.841± 0.018** | **0.911 ± 0.021** | **0.785 ± 0.023** |

---

> ### Author Response · Authors · 2024-11-20
>
> Table 10 RSNA results
>
> | setting                  |                     | RSNA, density   |                   |                   | RSNA, birads   |                  |                   |
> | ------------------------ | ------------------- | --------------- | ----------------- | ----------------- | -------------- | ---------------- | ----------------- |
> |                          | Previous work       | ACC             | AUROC             | AUPRC             | ACC            | AUROC            | AUPRC             |
> | Multimodal ，Single Task | Mirai               | 0.763±0.029     | 0.824 ± 0.026     | 0.635 ± 0.039     | 0.623±0.020    | 0.682 ± 0.022    | 0.553 ± 0.02      |
> |                          | Asymirai            | 0.741±0.014     | 0.810 ± 0.022     | 0.617 ± 0.030     | 0.601±0.031    | 0.670 ± 0.032    | 0.536 ± 0.035     |
> |                          | M4oE without MSOE   | 0.768±0.048     | 0.832± 0.042      | 0.642± 0.04       | 0.622±0.024    | 0.68 ± 0.019     | 0.550 ± 0.023     |
> |                          | M4oE                | 0.775±0.022     | 0.838 ± 0.016     | 0.644 ± 0.035     | 0.640±0.017    | 0.700 ± 0.013    | 0.576 ± 0.017     |
> | Multimodal Multi task    | Mirai               | 0.761±0.014     | 0.821 ± 0.02      | 0.645 ± 0.026     | 0.625±0.024    | 0.687 ± 0.022    | 0.565 ± 0.024     |
> | (medical ai)             | Mirai+MToE          | 0.768±0.009     | 0.834 ± 0.018     | 0.648 ± 0.03      | 0.631±0.027    | 0.689 ± 0.035    | 0.572 ± 0.020     |
> |                          | Asymirai            | 0.739±0.02      | 0.805 ± 0.016     | 0.638 ± 0.024     | 0.607±0.024    | 0.674 ± 0.03     | 0.555 ± 0.027     |
> |                          | Asymirai+MToE       | 0.742±0.012     | 0.816 ± 0.02      | 0.645 ± 0.041     | 0.612±0.025    | 0.686 ± 0.027    | 0.568 ± 0.023     |
> | Multimodal Multi-task    | EVIF                | 0.766±0.014     | 0.835 ± 0.024     | 0.674 ± 0.030     | 0.659±0.036    | 0.712 ± 0.033    | 0.596 ± 0.03      |
> | (natural domain)         | Fuller              | 0.769±0.0016    | 0.829 ± 0.014     | 0.673 ± 0.021     | 0.654±0.029    | 0.713 ± 0.025    | 0.597 ± 0.027     |
> |                          | AIDE                | 0.768±0.022     | 0.826 ± 0.023     | 0.672 ± 0.02      | 0.661±0.021    | 0.709 ± 0.02     | 0.589 ± 0.019     |
> |                          | MModN               | 0.771±0.018     | 0.834 ± 0.02      | 0.675 ± 0.025     | 0.664±0.023    | 0.718 ± 0.026    | 0.595 ± 0.026     |
> |                          | Multimodal Soft Moe | 0.762± 0.02     | 0.821± 0.024      | 0.646± 0.027      | 0.638±0.031    | 0.691±0.025      | 0.575±0.029       |
> |                          | AdaMV-MoE           | 0.767± 0.018    | 0.824± 0.022      | 0.67± 0.03        | 0.655±0.019    | 0.706±0.021      | 0.584±0.02        |
> |                          | Fuse-MoE            | 0.771± 0.024    | 0.834± 0.026      | 0.678± 0.037      | 0.663±0.027    | 0.714±0.032      | 0.592±0.035       |
> |                          | Ours without MTOE   | 0.767±0.022     | 0.823 ± 0.014     | 0.671 ± 0.024     | 0.643±0.027    | 0.701 ± 0.024    | 0.579 ± 0.021     |
> |                          | Ours                | **0.778±0.012** | **0.842 ± 0.015** | **0.682 ± 0.022** | **0.667±0.02** | **0.72 ± 0.017** | **0.601 ± 0.015** |

---

> ### Author Response · Authors · 2024-11-20
>
> Table 12 GAMMA Results
>
> |                          |                     | Glaucoma 3-class |                   |                   |
> | ------------------------ | ------------------- | ---------------- | ----------------- | ----------------- |
> |                          |                     | ACC              | AUROC             | AUPRC             |
> | Multimodal ，Single Task | Eyemost             | 0.860±0.017      | 0.910 ± 0.018     | 0.851 ± 0.022     |
> |                          | Eyestar             | 0.854±0.029      | 0.906 ± 0.022     | 0.841 ± 0.032     |
> |                          | M4oE without MSOE   | 0.862±0.018      | 0.912 ± 0.026     | 0.854 ± 0.025     |
> |                          | M4oE                | 0.876±0.025      | 0.927 ± 0.041     | 0.865 ± 0.026     |
> | Multimodal Multi task    | Eyemost             | 0.865±0.016      | 0.921 ± 0.022     | 0.858 ± 0.027     |
> | (medical ai)             | Eyemost+MToE        | 0.872±0.014      | 0.924 ± 0.014     | 0.861 ± 0.008     |
> |                          | Eyestar             | 0.859±0.022      | 0.899 ± 0.032     | 0.845 ± 0.033     |
> |                          | Eyestar+MToE        | 0.875±0.013      | 0.926 ± 0.008     | 0.865 ± 0.011     |
> | Multimodal Multi-task    | EVIF                | 0.887±0.017      | 0.936 ± 0.018     | 0.884 ± 0.016     |
> | (natural domain)         | Fuller              | 0.878±0.021      | 0.927 ± 0.019     | 0.877 ± 0.022     |
> |                          | AIDE                | 0.885±0.029      | 0.933 ± 0.025     | 0.881 ± 0.030     |
> |                          | MModN               | 0.892±0.016      | 0.943 ± 0.013     | 0.885 ± 0.014     |
> |                          | Multimodal Soft Moe | 0.871±0.029      | 0.918±0.027       | 0.863±0.025       |
> |                          | AdaMV-MoE           | 0.874±0.028      | 0.923±0.026       | 0.860±0.021       |
> |                          | Fuse-MoE            | 0.886±0.020      | 0.934±0.018       | 0.883±0.019       |
> |                          | Ours without MTOE   | 0.881±0.022      | 0.93 ± 0.039      | 0.879 ± 0.026     |
> |                          | Ours                | **0.904±0.017**  | **0.952 ± 0.015** | **0.895 ± 0.018** |
>
> Table 13 MIMIC Results
>
> | Multimodal MIMIC EXPERIMENTS |                             |                             |                      |                      |
> | ---------------------------- | --------------------------- | --------------------------- | -------------------- | -------------------- |
> |                              | In-hospital Mortality AUROC | In-hospital Mortality AUPRC | Length of Stay AUROC | Length of Stay AUPRC |
> | HAIM                         | 0.8091±0.054                | 0.4689±0.063                | 0.8174±0.035         | 0.7598±0.047         |
> | EVIF                         | 0.8179±0.046                | 0.5194±0.055                | 0.8156±0.037         | 0.7615±0.041         |
> | MModN                        | 0.8143±0.031                | 0.5225±0.039                | 0.8242±0.032         | 0.7777±0.028         |
> | Fuse-MoE                     | **0.8331±0.033**            | **0.5421±0.034**            | 0.8325±0.026         | 0.7844±0.024         |
> | Ours                         | <u>0.8310±0.039</u>         | <u>0.5368±0.041</u>         | **0.8561±0.025**     | **0.7891±0.037**     |

---

> ### Author Response · Authors · 2024-11-20
>
> Table 14 Subgroup Performance on Embeds Risk Prediction
>
> |          | Age   |       |       | Ethnicity        |       |        | Avg   |
> | -------- | ----- | ----- | ----- | ---------------- | ----- | ------ | ----- |
> |          | <50   | 50-70 | >70   | African American | White | Others |       |
> | baseline | 0.861 | 0.831 | 0.806 | 0.823            | 0.891 | 0.785  | 0.833 |
> | ours     | 0.866 | 0.862 | 0.85  | 0.857            | 0.872 | 0.848  | 0.859 |

---

> ### Author Response · Authors · 2024-11-25
>
> Dear Reviewer dm8E,
>
> Thank you very much for your valuable feedback. We have provided comprehensive point-by-point responses to address each of your concerns and questions. As we approach the end of the discussion period, we would greatly appreciate your feedback on whether our responses have adequately addressed your points. We remain available to provide any additional clarification you may need.
>
> Best regards,
>
> Submission 11923 Authors

---

> ### Comment · Reviewer_dm8E · 2024-11-26
>
> Thank you for quickly incorporating the feedback to my questions and concerns. Some of my concerns have been addressed, so I am happy to lift my score to a 5. However, I still believe that there is a fundamental problem of the paper claiming more than it validates with its experiments, as implied in my initial review. This prevents me from further increasing my score. Additional concerns that remain are:
>
> - W1: the definition of sample-dynamic fusion remains vague in the abstract and the terminology is inconsistent with “sample-adaptive” (L86). It remains unclear to me how this varies from any other multimodal fusion problem.
> - W3: Please introduce this in the abstract “[…] many significant medical imaging problems”. The three references you provided [11, 12, 13] even narrow this scope in the title which I believe would be suitable here, too.
> - Q3: While in a perfect world “different modalities should always carry some useful information” - in practice, this just isn’t the case. That’s why some experiments about these edge cases would strengthen the argument. This concern has not been adequately addressed in the Author response.

---

> > ### Author Response · Authors · 2024-11-28
> >
> > Hi Reviewer dm8E,
> >
> >   We highly appreciate your prompt response and thoughtful advice. We have carefully addressed your concerns through revisions and additional analyses:
> >
> > **W1: The definition of sample-dynamic fusion remains vague in the abstract and the terminology is inconsistent with “sample-adaptive” (L86). It remains unclear to me how this varies from any other multimodal fusion problem.**
> >
> > Response:
> >
> > In our paper, we use both “sample-adaptive” and “sample-dynamic fusion” to indicate the same “dynamic fusion” phenomenon in medical (imaging) problems, where the quality and amount of task-related information from different modalities vary across different patient samples due to factors like patient subgroups. To avoid confusion, we’ve unified this term as “sample-dynamic fusion/ dynamic fusion” across the whole manuscript, including the abstract and introduction. The changes are highlighted in blue.
> >
> > To clarify the distinction from conventional multimodal fusion: While traditional approaches typically combine modalities in a fixed manner, our dynamic fusion approach recognizes that the relative importance of different modalities can vary significantly across patient subgroups. Our method successfully addresses this by modeling the combination of modality-specific and shared information with dynamic soft MoE. Its effectiveness is demonstrated not only by the main experiments and ablation studies, but also by the subgroup analysis included in the previous response (Appendix Table 14).
> >
> > We have updated our manuscript for a clearer definition. Changes are marked blue in the Abstract Lines 15-24.
> >
> > To better understand the background of dynamic fusion, let’s revisit the analogy to infrared-RGB fusion:
> > Non-dynamic fusion ignores the dynamic changes in reality (e.g., Daylight vs Nighttime), which diminishes the visible texture in good lighting conditions and the infrared contrast in low lighting conditions[5].
> >
> > As for medical setting: Taking breast cancer screening as an example, non-dynamic fusion ignores the dynamic changes in reality (e.g., Younger vs Senior patients), which diminishes the FFDM’s advantage of seeing high-resolution details on less-dense breasts and 2DS’s reduced tissue overlap for denser breasts[8].
> >
> >
> >
> > **W3: Please introduce this in the abstract “[…] many significant medical imaging problems”. The three references you provided [11, 12, 13] even narrow this scope in the title which I believe would be suitable here, too.**
> >
> > Response:
> >
> > To make the definition and the scope to be more clear, we have taken the suggestion from the reviewer to further specify our main focus of medical imaging in the abstract and introduction sections. Along with this, the references  [11, 12, 13] are added in the first sentence of the introduction. Changes are marked blue in the revised manuscripts.

---

> > ### Author Response · Authors · 2024-11-28
> >
> > **Q3: While in a perfect world “different modalities should always carry some useful information” - in practice, this just isn’t the case. That’s why some experiments about these edge cases would strengthen the argument. This concern has not been adequately addressed in the Author response.**
> >
> > Response:
> >
> > Thank you for your suggestions! Your advice has been tremendously valuable and has helped us enhance the paper's contribution.
> >
> > In our previous response to Q3, we demonstrated that our model doesn’t uniformly assign near-equal modality utilization for every sample, but instead assigns varying importance to different modalities dynamically, based on both modality quality and task requirements. Whilst our experiment on EMBED yielded appearing even global modality contributions, the local modality contributions showed marked variation (Figure 5). In practice, we expect this pattern ought to differ across datasets and tasks.
> > To further explore this behaviour, as suggested by the reviewer, we have conducted additional edge-case experiments to explicitly examine how our model manages severely compromised modalities.
> >
> > These experiments utilised the RSNA dataset. We add Gaussian noise corruptions of 4 * X$\sim$N(0, 1) and 2 * X$\sim$N(0,1) to simulate practical scenarios with very noisy or corrupted modalities. For example, medical imaging modalities might be compromised or have little task-related information, due to poor acquisition or incorrect prescription. We maintained intact quality for both modalities in 70% of samples, whilst randomly corrupting one modality in the remaining 30%.
> >
> > The appendix Figure 10 contains examples showing the pre- and post-corruption images, which demonstrate the compromised modality settings as suggested by the reviewer. We have attached the results below and in the appendix Table 19. We observe that our model is well capable of utilising the non-corrupted modalities. Interestingly, we also see that while the edge cases cause modality utilization to be highly biased towards the intact modalities, the overall contribution remains close due to data distribution. This could potentially help to support our previous observation of the EMBED dataset, which shows close-to-even modality contributions in the global statistics.

---

> > ### Author Response · Authors · 2024-11-28
> >
> > Table 19: Edge Case (Modality Corruption Analysis) on RSNA Dataset
> >
> > |                                 |                              | Corruption with Gaussian Noise 4 * N(0,1)  |                          |                          |
> > | ------------------------------- | ---------------------------- | ------------------------------------------ | ------------------------ | ------------------------ |
> > |                                 | Overall Test                 | Subgroup Test                              |                          |                          |
> > |                                 | All (100%= 70%+15%+15%)      | 70% Uncorrupted Data                       | 15% corrupted Modality A | 15% corrupted Modality B |
> > | Density - Accuracy              | 0.761                        | 0.779                                      | 0.712                    | 0.727                    |
> > | BI-RADS - Accuracy              | 0.645                        | 0.666                                      | 0.608                    | 0.584                    |
> > | Density - Modality Contribution | [0.537,0.463]                | [ 0.539,0.461]                             | [0.195,0.805]            | [ 0.868,0.132]           |
> > | BI-RADS - Modality Contribution | [0.425,0.575]                | [0.408,0.592]                              | [0.157, 0.843]           | [ 0.773,0.227]           |
> > |                                 |                              |                                            |                          |                          |
> > |                                 | 100% Both Modality Corrupted |                                            |                          |                          |
> > | Density - Accuracy              | 0.453                        |                                            |                          |                          |
> > | BI-RADS - Accuracy              | 0.418                        |                                            |                          |                          |
> > |                                 |                              |                                            |                          |                          |
> > |                                 |                              | **Corruption with Gaussian Noise 2 * N(0, 1)** |                          |                          |
> > |                                 | Overall Test                 | Subgroup Test                              |                          |                          |
> > |                                 | All (100%= 70%+15%+15%)      | 70% Uncorrupted Data                       | 15% corrupted Modality A | 15% corrupted Modality B |
> > | Density - Accuracy              | 0.772                        | 0.782                                      | 0.736                    | 0.744                    |
> > | BI-RADS - Accuracy              | 0.652                        | 0.667                                      | 0.615                    | 0.603                    |
> > | Density - Modality Contribution | [0.538,0.462]                | [0.553,0.447]                              | [0.329, 0.671]           | [0.679,0.321]            |
> > | BI-RADS - Modality Contribution | [0.489, 0.511]               | [0.485,0.515]                              | [0.348, 0.652]           | [0.646,0.354 ]           |
> > |                                 | 100% Both Modality Corrupted |                                            |                          |                          |
> > | Density - Accuracy              | 0.664                        |                                            |                          |                          |
> > | BI-RADS - Accuracy              | 0.535                        |                                            |                          |                          |

---

> > > ### Comment · Reviewer_dm8E · 2024-12-02
> > >
> > > Thank you very much for going the extra mile and running additional experiments. More detail below:
> > >
> > > - W1: I cannot agree with the statement “While traditional approaches typically combine modalities in a fixed manner, our dynamic fusion approach recognizes that the relative importance of different modalities can vary significantly across patient subgroups.” - This statement is not true. Most fusion methods learn fusion operators which are dynamic - whether that is attention gating, bilinear fusion, or simply fitting an MLP on top of the concatenated embeddings. All of these methods are “dynamic” and put assign different weights to different modalities depending on the sample. The only “static” case would be if the fusion method is a fixed weighted function, which is rarely the case. I therefore believe that the “sample-adaptive” term just adds more confusion than clarity and overformalizes a relatively common problem in multimodal fusion. I would strongly encourage the authors to rectify this claim/terminology.
> > > - W3: Thank you for running the extra experiments. In clinical practice, I believe that this robustness to noise is a very interesting part (in my subjective view one of the most useful aspects of balancing the modality utilization through the MoEs). I would encourage the authors to promote this finding and investigate it in more depth.
> > >
> > > Nonetheless, I think that the response to W3 warrants an increase of the score to a 6 which will be my final score.

---

> > > > ### Author Response · Authors · 2024-12-03
> > > > **Thank you note**
> > > >
> > > > Dear Reviewer dm8E,
> > > >
> > > >   Thank you so much for your fast response and we want to say that your comments have added great contributions to this project! We will definitely highlight the exploration of noisy data scenarios. We will further enhance the manuscript in terms of dynamic fusion definition with support of references in our final version. MoEs are generally good at dynamic fusion because they create input-dependent data pathways through different experts, in other words, the weights are different. Our work further enhances this by studying modality domination and addressing it with the combination of MSoE and MToE modules.
> > > >
> > > >   Thank you again for your valuable feedback throughout this review process and raising the review score.
> > > >
> > > > Best Regards,
> > > >
> > > > Authors of Submission 11923

---

### Official Review · Reviewer_xATa · 2024-11-04

**Soundness:** 3
**Presentation:** 4
**Contribution:** 3
**Rating:** 6
**Confidence:** 3

**Summary:**

The paper mainly addresses two challenges in clinical tasks: patient-level and task-level dynamic fusion. For the patient-level fusion, a modality-specific MoE is employed. For the task-level fusion, a modality-task MoE with conditional MI regularization between experts and modalities given tasks is adopted. The experiments using EMBED, RSNA, VinDR, and GAMMA datasets outperform existing methods in both single-task and multi-task settings.

**Strengths:**

- The paper is well-structured and easy to follow.
- The motivation is clearly stated and convincing.
- The experiments show promising results over many baselines both in stand-alone and add-on manners.
- The paper adopted PID to make a fair comparison of synergy information.

**Weaknesses:**

- As far as I understand, there have been works leveraging the shared and specific information across modalities and should be included in discussions, see [1-3].
- Is there an ablation study for a reduced number of experts? How sensitive is this method when the number of experts decreases compared to other MoE methods? What is the procedure for choosing the number of experts?
- Please discuss the computational cost compared to the baselines.


[1]Wang, Hu, et al. "Multi-modal learning with missing modality via shared-specific feature modelling." Proceedings of the IEEE/CVF Conference on Computer Vision and Pattern Recognition. 2023.

[2]Yao, Wenfang, et al. "DrFuse: Learning Disentangled Representation for Clinical Multi-Modal Fusion with Missing Modality and Modal Inconsistency." Proceedings of the AAAI Conference on Artificial Intelligence. Vol. 38. No. 15. 2024.

[3]Chen, Cheng, et al. "Robust multimodal brain tumor segmentation via feature disentanglement and gated fusion." Medical Image Computing and Computer Assisted Intervention–MICCAI 2019: 22nd International Conference, Shenzhen, China, October 13–17, 2019, Proceedings, Part III 22. Springer International Publishing, 2019.

**Questions:**

- Are there any ablation studies on datasets other than EMBED?
- Are there any theoretical explanations on why the method mitigates gradient conflict?
- Is there any clinical interpretation of the results in Figure 5? For example, the difference in modality contribution across different tasks.

---

> ### Author Response · Authors · 2024-11-20
>
> We thank the reviewer for their valuable feedback. We updated our manuscript with some changes in writing and all additional results in Appendix Section F. Changes are marked in blue.
>
> **W1: “Discussion of related works”**
>
> R1: Thanks for the constructive feedback and suggestions for these papers! We agree that these papers can be relevant to our work in the perspective of multimodal fusion and feature disentangling. But our work provides a unique insight for dynamic modality-task modeling parametrized by MoE structure and dynamic sample fusion. We also provide sample-level and population-level interpretability.
> We have added these discussions with these mentioned works in our related works section (Sec. 2) as highlighted in the updated revision. We additionally include a recent NIPS’24 work that extends multimodal fusion for generative tasks.
>
> **W2: “Ablation study for number of experts”**
>
> R2: Yes, in original manuscript, we did conduct an ablation study about the number of experts as shown in Figure 7 as well as Section 4.4. From this ablation study, it shows generally more experts lead to better performance, in almost a linear relationship on two tasks’ performance.
>
> In order to investigate the sensitivity of our method on the number of experts, we also conduct an additional experiment: we removed all new task-related and modality-specific components proposed in our M4oE model to construct a basic version of multimodal MoE for comparison. The results are shown in Table 18. Compared to this basic MoE version, our method is less sensitive in the number of experts.
>
> Table 18:
>
> | n    | methoud      | risk1 | risk2 | risk3 | risk4 | risk5 | density |
> | ---- | ------------ | ----- | ----- | ----- | ----- | ----- | ------- |
> | 16   | ours         | 83.7  | 73.8  | 72.5  | 70.9  | 72.4  | 82.1    |
> | 32   | ours         | 84.4  | 74.1  | 72.8  | 71.5  | 72.8  | 82.5    |
> | 64   | ours         | 85    | 74.9  | 73.2  | 72    | 73.3  | 83.7    |
> | 128  | ours         | 85.9  | 75.5  | 73.7  | 72.7  | 73.7  | 84.1    |
> | 16   | baseline moe | 80.9  | 71.7  | 71.2  | 69.5  | 70.6  | 81.3    |
> | 32   | baseline moe | 82.1  | 72.5  | 71.7  | 70.4  | 71.4  | 82      |
> | 64   | baseline moe | 83    | 73.2  | 72.3  | 71.2  | 72    | 82.7    |
> | 128  | baseline moe | 83.8  | 73.7  | 72.9  | 72    | 72.4  | 83.3    |
>
> **W3: “Computational cost”**
>
> R3: We report the computational cost of our model (inference time, training time, parameters) in Table 16. Note this analysis is conducted on a training batch size of 32 on the RSNA dataset. In the implementation, we used a single A100 80G GPU with 8 cores CPU to train the model. By comparing with other methods, our approach remains a similar time efficiency without adding latency, while using a similar scale of model parameters as other MoE methods. This is because the backbone of MoE introduces more model parameters than other structures [4].
>
> MoE architectures in our method provide a promising way to scale the model size without paying too much computational cost. Our backbone, the recently introduced soft-MoE, runs at a faster speed than regular ViTs with 10x trainable parameters[4]. Similarly, M4oE can also scale to larger model sizes and datasets without introducing tremendous computational complexity.
>
>
> Table 16
> | Complexity           | mirai  | asymmirai | EVIF   | Fuller | AIDE   | Ours | Fuse-MoE |
> | -------------------- | ------ | --------- | ------ | ------ | ------ | ---- | -------- |
> | inference time/batch | 0.37   | 0.38      | 0.5    | 0.35   | 0.36   | 0.47 | 0.44     |
> | training time/batch  | 1.15   | 1.01      | 1.56   | 0.98   | 1.12   | 1.51 | 1.32     |
> | params               | 120.4M | 205.4M    | 188.1M | 211.6M | 254.3M | 3.2B | 2.1B     |

---

> ### Author Response · Authors · 2024-11-20
>
> **Q1: “Ablation studies on more datasets”**
>
> R4: Besides EMBED, we have conducted ablation studies on more datasets including GAMMA, RSNA, and VinDR datasets. The results are shown in Table 17.
> We observe that: In datasets where the target tasks are more different(GAMMA), MToE and conditional MI loss are slightly more important.
> In datasets with higher modality discrepancy (MIMIC), MSoE plays a more important role.
>
>
> | Ablation study |      |        | EMBED |         |        | RSNA    |        | VinDR   |        | GAMMA            |                          |
> | ----------------------- | ---- | ------ | ----- | ------- | ------ | ------- | ------ | ------- | ------ | ---------------- | ------------------------ |
> | Task MoE                | MSoE | MI Reg | risk  | density | birads | density | birads | density | birads | Glaucoma 3-class | Optical Cup Segmentation |
> | no                      |      | no     | 84    | 82.8    | 73.1   | 76.7    | 64.3   | 86.9    | 67.3   | 88.1             | 87.4                     |
> |                         | no   |        | 85.1  | 83.5    | 73.8   | 77.1    | 65.9   | 88.3    | 68.7   | 89.5             | 87                       |
> |                         |      | no     | 85.2  | 83.7    | 73.5   | 77.5    | 66.4   | 89.1    | 70.6   | 90               | 88.9                     |
> |                         |      |        | 85.9  | 84.1    | 75.1   | 77.8    | 66.7   | 89.6    | 71.8   | 90.4             | 89.7                     |
>
>
> **Q2: “Theoretical explanations on mitigating gradient conflict”**
>
> R5: Gradient conflicts in multi-task learning with MoE often result from assigning counteracting tasks to the same expert [3].  As theoretically analyzed in [1], incorporating sparse task-specific modules helps learn the disentangled features of different tasks. This task specialization can help mitigate the counteracting gradient on experts in the MoE blocks. Therefore, our task MoE module uses separate task slots and task-specific embeddings to encourage separate feature learning.
>
> Many existing MoE works [4,5] apply the well-known load-balancing loss. One outcome of this loss is that it pushes expert usage across tasks and batches to be similar. However, this may not be helpful for multi-task learning since experts may be forced to share among tasks with conflicting gradients. Our loss encourages stronger dependence between experts and tasks. The analysis of task-expert sparsity in Figure 6 (b) and (c) shows that our method achieves sparser expert-task relationships to facilitate task specialization, so that it can help to alleviate the gradient conflict problem.
>
> **Q3: Clinical interpretation of Fig. 5**
>
> R6:
>
> In the sample-level:
>
> Patient 1: Dense glandular tissue spread throughout the breast. High breast density can obscure lesions in traditional mammography, but 2DS images provide improved lesion visibility in dense breasts [6]. According to [7], women younger than 50 years, who usually have dense breasts, may benefit more from 2DS than from FFDM alone [7]. Therefore, it aligns with our observation that M4oE utilizes 2DS much more than FFDM for risk predictions. This indicates a positive sign that the model does not rely heavily on FFDM and provides a potential way for early risk detection[8].
>
> Patient 2: Primarily fatty tissue with less dense glandular areas makes this patient easier to evaluate for abnormalities. Usually, for this patient subgroup, FFDM alone often suffices for accurate screening [9]. This also matches our observation that the model utilizes FFDM more than 2DS for risk prediction task.
>
> In the population-level:
>
> A combination of both FFDM and 2DS is proven to be highly effective on different populations[10,11]. Our model utilizes all of the modalities well for the predictions, while FFDM has indicated slightly higher importance than 2DS on average. We suspect this may be attributed to the relatively higher age distribution (mean=58.5 y) in the dataset.

---

> ### Author Response · Authors · 2024-11-20
>
> [1] Lachapelle S, Deleu T, Mahajan D, et al. Synergies between disentanglement and sparsity: Generalization and identifiability in multi-task learning[C]//International Conference on Machine Learning. PMLR, 2023: 18171-18206.
>
> [2] Yu T, Kumar S, Gupta A, et al. Gradient surgery for multi-task learning[J]. Advances in Neural Information Processing Systems, 2020, 33: 5824-5836.
>
> [3] Chen Z, Shen Y, Ding M, et al. Mod-squad: Designing mixtures of experts as modular multi-task learners[C]//Proceedings of the IEEE/CVF Conference on Computer Vision and Pattern Recognition. 2023: 11828-11837.
>
> [4] Riquelme C, Puigcerver J, Mustafa B, et al. Scaling vision with sparse mixture of experts[J]. Advances in Neural Information Processing Systems, 2021, 34: 8583-8595.
>
> [5] Fan Z, Sarkar R, Jiang Z, et al. M³vit: Mixture-of-experts vision transformer for efficient multi-task learning with model-accelerator co-design[J]. Advances in Neural Information Processing Systems, 2022, 35: 28441-28457.
>
> [6] Aujero, M. P., Gavenonis, S. C., Benjamin, R., Zhang, Z., & Holt, J. S. (2017). Clinical performance of synthesized two-dimensional mammography combined with tomosynthesis in a large screening population. Radiology, 283(1), 70-76.
>
> [7] McDonald, E. S., McCarthy, A. M., Akhtar, A. L., Synnestvedt, M. B., Schnall, M., & Conant, E. F. (2015). Baseline screening mammography: performance of full-field digital mammography versus digital breast tomosynthesis. American Journal of Roentgenology, 205(5), 1143-1148.
>
> [8] Harbeck, N., Penault-Llorca, F., Cortes, J., Gnant, M., Houssami, N., Poortmans, P., ... & Cardoso, F. (2019). Breast cancer. Nature reviews Disease primers, 5(1), 66.
>
> [9] Destounis, S. V., Santacroce, A., & Arieno, A. (2020). Update on breast density, risk estimation, and supplemental screening. American Journal of Roentgenology, 214(2), 296-305.
>
> [10] Mumin, N. A., Rahmat, K., Fadzli, F., Ramli, M. T., Westerhout, C. J., Ramli, N., ... & Ng, K. H. (2019). Diagnostic efficacy of synthesized 2D digital breast tomosynthesis in multi-ethnic Malaysian population. Scientific Reports, 9(1), 1459.
>
> [11] Nakajima, E., Tsunoda, H., Ookura, M., Ban, K., Kawaguchi, Y., Inagaki, M., ... & Ishikawa, T. (2021). Digital breast tomosynthesis complements two-dimensional synthetic mammography for secondary examination of breast cancer. Journal of the Belgian Society of Radiology, 105(1).

---

> ### Author Response · Authors · 2024-11-25
>
> Dear Reviewer xATa,
>
> Thank you very much for your valuable feedback. We have provided comprehensive point-by-point responses to address each of your concerns and questions. As we approach the end of the discussion period, we would greatly appreciate your feedback on whether our responses have adequately addressed your points. We remain available to provide any additional clarification you may need.
>
> Best regards,
>
> Submission 11923 Authors

---

> > ### Comment · Reviewer_xATa · 2024-11-26
> >
> > Thanks to the authors for providing additional information. I stand by my overall evaluation and thank the authors for their clarification.

---

> > > ### Author Response · Authors · 2024-12-03
> > >
> > > Dear Reviewer xATa,
> > >
> > > Thank you again for your thoughtful review and prompt response to our rebuttal. Please feel free to let us know if you have any remaining concerns. We welcome any additional feedback or questions you may have.
> > >
> > > Thank you for your time and consideration.
> > >
> > > Best Regards,
> > >
> > > Authors of Submission 11923

---

### Official Review · Reviewer_P7K2 · 2024-11-04

**Soundness:** 3
**Presentation:** 2
**Contribution:** 3
**Rating:** 5
**Confidence:** 4

**Summary:**

In this paper, the author present a new framework for training multi-modal networks, called the Multi-modal Multi-task Mixture of Experts. The framework consists of two components:
- MSoE: Modality specific mixture of experts --> for each modality, they learn a function g that applies:
 column-wise softmax (D) on X times a learnable matrix, multiplied by X, followed by row-wise softmax (C) on the output and a linear combination to compute the prediction.
- MToE: Modality shared modality task mixture of experts --> connects tasks to input modalities by learning a task embedding shared across experts.

They also propose a mutual information loss and evaluate the approach on four publicly available medical imaging datasets for breast cancer and OCT.

**Strengths:**

The framework presented is original and interesting. It outperforms existing baseline models. The authors run experiments on multiple datasets and conduct an ablation study.

**Weaknesses:**

- The paper presentation requires improvement. For example, there is unnecessary use of ; and there is incorrect use of opening quotations ". The authors also repeatedly introduce the abbreviations - this should be done once.
- I found it difficult to parse through Figure 2 (Can you relate it with the textual explanation of the functions?)
- The authors only compare to a few baselines, can you incorporate more? There is a lot of literature on multimodal learning now.
- Are the performance improvements significant? Can you conduct significance testing and provide confidence intervals?
- The experiments are conducted on medical imaging datasets. How does this apply to other non-imaging modalities where modality competition may be more pronounced. For example, this could be applicable to MIMIC CXR (chest X-rays) and MIMIC EHR where downstream tasks are more dependent on the EHR modality.
- The main results section in the text should also discuss the quantitative results.
- What was your hyperparameter tuning strategy? It is unclear if these baselines have been best optimized.
- Can you also compute AUROC and AUPRC for the classification tasks? Accuracy is not sufficient.

**Questions:**

- Can the authors discuss the scalability of the framework? What is the computational complexity?

---

> ### Author Response · Authors · 2024-11-20
>
> We thank the reviewer for their valuable feedback. We updated our manuscript with some changes in writing and all additional results in Appendix Section F. Changes are marked in blue.
>
> **W1: “Writing improvement”**
>
> R1: Thanks for pointing this out! We have revised the manuscript carefully to address these issues in the updated revision.
>
> **W2: “Figure 2”**
>
> R2: Figure 2 primarily corresponds to Section 3.2, where we provide a comprehensive overview of the framework and the functionality of each component. A detailed description of the data flow and the explanation of different modules is presented in original paper Section 3.2. Here is a high-level explanation to help with interpretation:
>
> Our framework comprises four components: (1) modality-specific feature embedders, (2) modality-specific Soft MoEs (MSoE), (3) Modality-Task Soft MoEs (MToE), and (4) a final fusion block. Each module is described in more details below.
> The framework first embeds different input modalities with feature embedders, and then projects m input modalities to embeddings $X_1,..., X_m \in \mathbb{R}^{l \times d}$. This is shown in Fig 2-a) part 1.
>
> 1) To mitigate the modality competition phenomenon, for each modality $M_i$ we use MSoE blocks to learn and retain the modality-specific features $Z_i \in \mathbb{R}^{l \times d}$ from $X_i$. Detailed MSoE is in Fig 2-b), and data flow is in Fig 2-a) part 2.
>
> 2) To model shared modality information and modality-task dependence, we feed $X_1, \dots, X_m$ into MToE blocks and obtain the task-dependent fused modality features $G_1, \dots, G_p $. MToE uses learnable task embeddings and task slots to facilitate the dynamic connection between modalities and tasks. This also allows for probability modeling with the $\phi$ matrix. Detailed MToE is in Fig 2-c), and data flow is in Fig 2-a) part 2.
>
> 3) Finally, MToE outputs from $G_1,...,G_p$, and MSoE outputs $Z_1,...,Z_m$ collectively form a pool $H_1,...,H_p$. For each task, $T_k$, $H_k$ is passed to the task heads after a basic soft MoE block for final predictions. The fusion block structure is Fig 2-d) and data flow in Fig 2-a) part 3.
> To make it more clear, we modify the caption of Fig. 2 and Sec. 3.2 to add the textual explanation of the functions (highlighted in blue).
>
> **W3: “More baseline comparison.”**
>
> R3: In our paper, we have conducted an extension baseline comparison, including  (1) two baselines targeting each medical applications domain in mammography, ophthalmology, radiology+EHR. They are also adapted to integrate with MToE to construct a stronger model for a fair comparison; (2)  we also compare with other 7 multimodal multitask baselines in the natural domain. The comparison in Table 9-13 of the original paper over 5 benchmarks has shown the solid improvement of our proposed approach.
>
> In order to compare against more baselines as suggested, we conduct more experiments and compare with more baselines which are all very recently released works (ICML’24 Multimodal Soft MoE[4],  ICCV’23 AdaMV-MoE [1], and NIPS’24 FuseMoE [2]).
> The experimental results are shown in Table 9-13, added in the revision. These additional results further show that our proposed method consistently performs well.
>
>
> **W4: “Result significance”**
>
> R4:  As suggested, we report the mean and standard deviation from five-fold cross-validation in the reported results in Table 9-13 on multiple metrics, including Accuracy, AUROC, and AUPRC.
>
> **W5: “Experiments on non-imaging modalities.”**
>
> R5: Since the proposed method mainly focuses on modeling the dynamic multimodal fusion for multi-task learning, we think our method is able to generalize to other modalities as well under this scheme. To demonstrate this, we performed additional multi-modal multi-task experiments on the MIMIC dataset following the experiment setting in [2][8]. Specifically, we used X-ray images, clinical/radiology notes, and electronic health records (EHR) modalities as inputs, to predict two tasks including in-hospital mortality prediction, and binary binned length of stay. The results are shown in Table 12.
> Through comparison with baseline methods, our method can achieve comparable results on mortality prediction and outperforms on length of stay prediction. Although this is not the main focus of our paper, we hope these additional experiments can convince that our proposed approach is potentially generalizable to non-imaging modalities as well.
>
> **W6: “Discuss the quantitative results.”**
>
> R6: We have already discussed the main conclusion from the experiment results in Section 4.2. Due to the page limit, we kept it succinct. As suggested by the reviewer, we have added more discussion based on the quantitative analysis in Sec. 4.2 in revision, as highlighted in blue.

---

> > ### Author Response · Authors · 2024-11-20
> >
> > **W7: “Hyperparameter tuning strategy.”**
> >
> > R7:
> > Ours: We determined optimal $\alpha$ weight for conditional mutual information loss and the number of experts $n$ based on the validation set conducted on the EMBED dataset. Then we keep using the same hyperparameters for our methods on all the benchmarks.
> >
> > For baselines, we followed their experimental sections and ablation analysis to set the best-performing hyperparameters. We also keep the learning rate =1e-4 and batch size =32 fixed for all the baseline methods, including our approach, for a fair comparison. The hyperparameters used for baseline methods are:
> >
> > MModN[7] : We set dropout rate = 0.1 and state representation size = 20.
> >
> > EVIF[3]: We set the weight $\gamma_1=1,\gamma_2 =0.1, \gamma_3=0.01$ for the task loss, mutual information minimization loss, and mutual information maximization loss respectively.
> >
> > AIDE[6]: Instead of the original 1e-3 learning rate, we changed it to 1e-4 to keep comparisons fair.
> >
> > Fuller[5]: We set the weight factor $\alpha$ between modalities in gradient calibration to be 0.1. The gradient momentum hyperparameter m is set to 0.2.
> >
> > AdaMV-MoE[1]: We set the auxiliary loss weights to be 5e−3 and delta_n to be 2000.
> >
> > We have added content about hyperparameter tuning and settings in Appendix Sec. C.3.5  to further clarify this clearly. See changes in revision highlighted in blue.
> >
> > **W8: “Compute AUROC and AUPRC”**
> >
> > R8: We have added the new metrics for our main experiments and reported AUROC and AUPRC in Appendix Tables 9-13. Along with the Accuracy, our proposed method achieves superior performance in different metrics.
> >
> > **Q1: “Computational complexity and framework scalability”**
> >
> > R9: We report the computational cost of our model (inference time, training time, parameters) in Table 16. Note this analysis is conducted on a training batch size of 32 on the RSNA dataset. In the implementation, we used a single A100 80G GPU with 8 cores CPU to train the model. By comparing with other methods, our approach remains a similar time efficiency without adding latency, while using a similar scale of model parameters as other MoE methods. This is because the backbone of MoE introduces more model parameters than other structures [4].
> >
> > MoE architectures in our method provide a promising way to scale the model size without paying too much computational cost. Our backbone, the recently introduced soft-MoE, runs at a faster speed than regular ViTs with 10x trainable parameters[4]. Similarly, M4oE can also scale to larger model sizes and datasets without introducing tremendous computational complexity.
> >
> >
> > [1] Chen T, Chen X, Du X, et al. Adamv-moe: Adaptive multi-task vision mixture-of-experts[C] //Proceedings of the IEEE/CVF International Conference on Computer Vision. 2023: 17346-17357.
> >
> > [2] Han X, Nguyen H, Harris C, et al. Fusemoe: Mixture-of-experts transformers for fleximodal fusion[J]. arXiv preprint arXiv:2402.03226, 2024.
> >
> > [3] Geng M, Zhu L, Wang L, et al. Event-based Visible and Infrared Fusion via Multi-task Collaboration[C]//Proceedings of the IEEE/CVF Conference on Computer Vision and Pattern Recognition. 2024: 26929-26939.
> >
> > [4] Puigcerver, J., Riquelme, C., Mustafa, B., & Houlsby, N. (2023). From sparse to soft mixtures of experts. arXiv preprint arXiv:2308.00951.
> >
> > [5] Huang Z, Lin S, Liu G, et al. Fuller: Unified multi-modality multi-task 3d perception via multi-level gradient calibration[C]//Proceedings of the IEEE/CVF International Conference on Computer Vision. 2023: 3502-3511.
> >
> > [6] Yang D, Huang S, Xu Z, et al. Aide: A vision-driven multi-view, multi-modal, multi-tasking dataset for assistive driving perception[C]//Proceedings of the IEEE/CVF International Conference on Computer Vision. 2023: 20459-20470.
> >
> > [7] Swamy, V., Satayeva, M., Frej, J., Bossy, T., Vogels, T., Jaggi, M., ... & Hartley, M. A. (2024). Multimodn—multimodal, multi-task, interpretable modular networks. Advances in Neural Information Processing Systems, 36.
> >
> > [8] Soenksen L R, Ma Y, Zeng C, et al. Integrated multimodal artificial intelligence framework for healthcare applications[J]. NPJ digital medicine, 2022, 5(1): 149.

---

> > > ### Author Response · Authors · 2024-11-20
> > > **Table 9**
> > >
> > > | Setting                  | Method              | EMBED, risk     |                   |                   | EMBED, birads   |                   |                   | EMBED, density   |                   |                   |
> > > | ------------------------ | ------------------- | --------------- | ----------------- | ----------------- | --------------- | ----------------- | ----------------- | ---------------- | ----------------- | ----------------- |
> > > |                          |                     | ACC             | AUROC             | AUPRC             | ACC             | AUROC             | AUPRC             | ACC              | AUROC             | AUPRC             |
> > > | Multimodal ，Single Task | Mirai               | 0.840±0.033     | 0.769 ± 0.039     | 0.653 ± 0.050     | 0.725±0.021     | 0.701 ± 0.014     | 0.598 ± 0.019     | 0.823± 0.030     | 0.898 ± 0.027     | 0.751 ± 0.034     |
> > > |                          | Asymirai            | 0.790±0.053     | 0.765 ± 0.022     | 0.647 ± 0.030     | 0.694±0.015     | 0.690 ± 0.018     | 0.580 ± 0.021     | 0.802± 0.029     | 0.875 ± 0.026     | 0.714 ± 0.03      |
> > > |                          | M4oE without MSOE   | 0.832±0.026     | 0.768 ± 0.014     | 0.650 ± 0.043     | 0.719±0.030     | 0.695 ± 0.041     | 0.585 ± 0.037     | 0.821± 0.019     | 0.898 ± 0.013     | 0.752 ± 0.026     |
> > > |                          | M4oE                | 0.855±0.041     | 0.791 ± 0.030     | 0.663 ± 0.034     | 0.742±0.027     | 0.705 ± 0.033     | 0.603 ± 0.028     | 0.836± 0.034     | 0.900 ± 0.039     | 0.768 ± 0.025     |
> > > | Multimodal Multi task    | Mirai               | 0.831±0.047     | 0.776 ± 0.054     | 0.662 ± 0.043     | 0.723±0.021     | 0.708 ± 0.019     | 0.605 ± 0.027     | 0.832± 0.026     | 0.894 ± 0.028     | 0.763 ± 0.021     |
> > > | (medical ai)             | Mirai+MToE          | 0.846±0.031     | 0.781 ± 0.050     | 0.671 ± 0.070     | 0.734±0.028     | 0.712 ± 0.031     | 0.607 ± 0.023     | 0.833± 0.030     | 0.897 ± 0.022     | 0.771 ± 0.029     |
> > > |                          | Asymirai            | 0.800±0.057     | 0.765 ± 0.034     | 0.654 ± 0.047     | 0.696±0.034     | 0.676 ± 0.026     | 0.589 ± 0.020     | 0.825± 0.035     | 0.877 ± 0.026     | 0.755 ± 0.040     |
> > > |                          | Asymirai+MToE       | 0.819±0.061     | 0.771 ± 0.059     | 0.665 ± 0.056     | 0.708±0.019     | 0.682 ± 0.030     | 0.602 ± 0.029     | 0.831± 0.027     | 0.889 ± 0.014     | 0.760 ± 0.021     |
> > > | Multimodal Multi-task    | EVIF                | 0.847±0.034     | 0.813 ± 0.038     | 0.682 ± 0.024     | 0.736±0.022     | 0.719± 0.035      | 0.619 ± 0.031     | 0.834± 0.019     | 0.906 ± 0.017     | 0.773 ± 0.023     |
> > > | (natural domain)         | Fuller              | 0.843±0.021     | 0.793 ± 0.035     | 0.681 ± 0.045     | 0.725±0.057     | 0.718 ± 0.047     | 0.622 ± 0.053     | 0.835± 0.033     | 0.909 ± 0.034     | 0.778 ± 0.038     |
> > > |                          | AIDE                | 0.841±0.018     | 0.790 ± 0.026     | 0.678 ± 0.026     | 0.736±0.032     | 0.721± 0.028      | 0.623± 0.035      | 0.829± 0.022     | 0.891 ± 0.020     | 0.771 ± 0.030     |
> > > |                          | MModN               | 0.848±0.022     | 0.815 ± 0.049     | 0.686 ± 0.065     | 0.739±0.020     | 0.722 ± 0.026     | 0.629 ± 0.019     | 0.837± 0.034     | 0.910 ± 0.032     | 0.776 ± 0.027     |
> > > |                          | Multimodal Soft Moe | 0.833±0.012     | 0.779±0.014       | 0.665±0.015       | 0.721±0.031     | 0.702±0.024       | 0.593±0.038       | 0.815±0.030      | 0.872±0.024       | 0.750±0.036       |
> > > |                          | AdaMV-MoE           | 0.845±0.021     | 0.810±0.023       | 0.679±0.029       | 0.729±0.024     | 0.717±0.027       | 0.624±0.023       | 0.828±0.039      | 0.887±0.037       | 0.762±0.031       |
> > > |                          | Fuse-MoE            | 0.847±0.030     | 0.817±0.042       | 0.684±0.028       | 0.740±0.028     | 0.724±0.019       | 0.631±0.027       | 0.835±0.036      | 0.909±0.032       | 0.774±0.028       |
> > > |                          | Ours without MTOE   | 0.840±0.021     | 0.813 ± 0.030     | 0.680 ± 0.059     | 0.731±0.021     | 0.711 ± 0.019     | 0.608 ± 0.024     | 0.828± 0.028     | 0.887± 0.025      | 0.764 ± 0.030     |
> > > |                          | Ours                | **0.859±0.023** | **0.831 ± 0.021** | **0.708 ± 0.024** | **0.751±0.024** | **0.739 ± 0.023** | **0.642 ± 0.025** | **0.841± 0.018** | **0.911 ± 0.021** | **0.785 ± 0.023** |

---

> ### Author Response · Authors · 2024-11-20
> **Table 10**
>
> | setting                  |                     | RSNA, density |                   |                   | RSNA, birads   |                  |                   |
> | ------------------------ | ------------------- | ------------------------------------------------------------ | ----------------- | ----------------- | -------------- | ---------------- | ----------------- |
> |                          | Previous work       | ACC                                                          | AUROC             | AUPRC             | ACC            | AUROC            | AUPRC             |
> | Multimodal ，Single Task | Mirai               | 0.763±0.029                                                  | 0.824 ± 0.026     | 0.635 ± 0.039     | 0.623±0.020    | 0.682 ± 0.022    | 0.553 ± 0.02      |
> |                          | Asymirai            | 0.741±0.014                                                  | 0.810 ± 0.022     | 0.617 ± 0.030     | 0.601±0.031    | 0.670 ± 0.032    | 0.536 ± 0.035     |
> |                          | M4oE without MSOE   | 0.768±0.048                                                  | 0.832± 0.042      | 0.642± 0.04       | 0.622±0.024    | 0.68 ± 0.019     | 0.550 ± 0.023     |
> |                          | M4oE                | 0.775±0.022                                                  | 0.838 ± 0.016     | 0.644 ± 0.035     | 0.640±0.017    | 0.700 ± 0.013    | 0.576 ± 0.017     |
> | Multimodal Multi task    | Mirai               | 0.761±0.014                                                  | 0.821 ± 0.02      | 0.645 ± 0.026     | 0.625±0.024    | 0.687 ± 0.022    | 0.565 ± 0.024     |
> | (medical ai)             | Mirai+MToE          | 0.768±0.009                                                  | 0.834 ± 0.018     | 0.648 ± 0.03      | 0.631±0.027    | 0.689 ± 0.035    | 0.572 ± 0.020     |
> |                          | Asymirai            | 0.739±0.02                                                   | 0.805 ± 0.016     | 0.638 ± 0.024     | 0.607±0.024    | 0.674 ± 0.03     | 0.555 ± 0.027     |
> |                          | Asymirai+MToE       | 0.742±0.012                                                  | 0.816 ± 0.02      | 0.645 ± 0.041     | 0.612±0.025    | 0.686 ± 0.027    | 0.568 ± 0.023     |
> | Multimodal Multi-task    | EVIF                | 0.766±0.014                                                  | 0.835 ± 0.024     | 0.674 ± 0.030     | 0.659±0.036    | 0.712 ± 0.033    | 0.596 ± 0.03      |
> | (natural domain)         | Fuller              | 0.769±0.0016                                                 | 0.829 ± 0.014     | 0.673 ± 0.021     | 0.654±0.029    | 0.713 ± 0.025    | 0.597 ± 0.027     |
> |                          | AIDE                | 0.768±0.022                                                  | 0.826 ± 0.023     | 0.672 ± 0.02      | 0.661±0.021    | 0.709 ± 0.02     | 0.589 ± 0.019     |
> |                          | MModN               | 0.771±0.018                                                  | 0.834 ± 0.02      | 0.675 ± 0.025     | 0.664±0.023    | 0.718 ± 0.026    | 0.595 ± 0.026     |
> |                          | Multimodal Soft Moe | 0.762± 0.02                                                  | 0.821± 0.024      | 0.646± 0.027      | 0.638±0.031    | 0.691±0.025      | 0.575±0.029       |
> |                          | AdaMV-MoE           | 0.767± 0.018                                                 | 0.824± 0.022      | 0.67± 0.03        | 0.655±0.019    | 0.706±0.021      | 0.584±0.02        |
> |                          | Fuse-MoE            | 0.771± 0.024                                                 | 0.834± 0.026      | 0.678± 0.037      | 0.663±0.027    | 0.714±0.032      | 0.592±0.035       |
> |                          | Ours without MTOE   | 0.767±0.022                                                  | 0.823 ± 0.014     | 0.671 ± 0.024     | 0.643±0.027    | 0.701 ± 0.024    | 0.579 ± 0.021     |
> |                          | Ours                | **0.778±0.012**                                              | **0.842 ± 0.015** | **0.682 ± 0.022** | **0.667±0.02** | **0.72 ± 0.017** | **0.601 ± 0.015** |

---

> ### Author Response · Authors · 2024-11-20
> **Table 11**
>
> | setting                  |                     | Vindr, density  |                   |                   | VinDR, birads   |                   |                   |
> | ------------------------ | ------------------- | --------------- | ----------------- | ----------------- | --------------- | ----------------- | ----------------- |
> |                          | Baselines           | ACC             | AUROC             | AUPRC             | ACC             | AUROC             | AUPRC             |
> | Multimodal ，Single Task | Mirai               | 0.863±0.025     | 0.824 ± 0.014     | 0.717 ± 0.011     | 0.661±0.023     | 0.705 ± 0.014     | 0.571 ± 0.010     |
> |                          | Asymirai            | 0.789±0.012     | 0.742 ± 0.032     | 0.645 ± 0.012     | 0.624±0.025     | 0.680 ± 0.016     | 0.537 ± 0.017     |
> |                          | M4oE without MSOE   | 0.854±0.015     | 0.814 ± 0.014     | 0.661 ± 0.016     | 0.658±0.017     | 0.703 ± 0.012     | 0.569 ± 0.016     |
> |                          | M4oE                | 0.877±0.019     | 0.857 ± 0.012     | 0.753 ± 0.020     | 0.675±0.019     | 0.714 ± 0.015     | 0.590 ± 0.017     |
> | Multimodal Multi task    | Mirai               | 0.854±0.017     | 0.822 ± 0.028     | 0.708 ± 0.018     | 0.659±0.014     | 0.702 ± 0.017     | 0.564 ± 0.016     |
> | (medical ai)             | Mirai+MToE          | 0.859±0.014     | 0.831 ± 0.013     | 0.714 ± 0.012     | 0.664±0.013     | 0.710 ± 0.016     | 0.579 ± 0.012     |
> |                          | Asymirai            | 0.791±0.015     | 0.752 ± 0.022     | 0.654 ± 0.011     | 0.622±0.025     | 0.671 ± 0.012     | 0.528 ± 0.018     |
> |                          | Asymirai+MToE       | 0.812±0.016     | 0.780 ± 0.016     | 0.667 ± 0.016     | 0.642±0.017     | 0.69 ± 0.015      | 0.552 ± 0.021     |
> | Multimodal Multi-task    | EVIF                | 0.888±0.025     | 0.869 ± 0.021     | 0.760 ± 0.019     | 0.707±0.021     | 0.734 ± 0.026     | 0.62 ± 0.019      |
> | (natural domain)         | Fuller              | 0.871±0.017     | 0.856 ± 0.015     | 0.75 ± 0.013      | 0.687±0.014     | 0.722 ± 0.035     | 0.602 ± 0.023     |
> |                          | AIDE                | 0.882±0.017     | 0.867 ± 0.017     | 0.755 ± 0.025     | 0.698±0.019     | 0.726 ± 0.016     | 0.609 ± 0.014     |
> |                          | MModN               | 0.890±0.018     | 0.878 ± 0.021     | 0.765 ± 0.023     | 0.704±0.017     | 0.732 ± 0.015     | 0.613 ± 0.018     |
> |                          | Multimodal Soft Moe | 0.849 ± 0.02    | 0.806 ±  0.019    | 0.664 ±  0.021    | 0.653 ± 0.022   | 0.7 ± 0.026       | 0.564 ±  0.019    |
> |                          | AdaMV-MoE           | 0.88 ± 0.016    | 0.861 ±  0.015    | 0.754 ±  0.018    | 0.679 ± 0.018   | 0.712 ±  0.016    | 0.595 ±  0.014    |
> |                          | Fuse-MoE            | 0.892 ± 0.028   | 0.881 ±  0.02     | 0.764 ±  0.017    | 0.705 ± 0.026   | 0.733 ±  0.018    | 0.618 ±  0.019    |
> |                          | Ours without MTOE   | 0.869±0.020     | 0.848 ± 0.018     | 0.744 ± 0.018     | 0.673±0.020     | 0.716 ± 0.019     | 0.588 ± 0.016     |
> |                          | Ours                | **0.896±0.017** | **0.888 ± 0.014** | **0.772 ± 0.016** | **0.718±0.015** | **0.739 ± 0.013** | **0.629 ± 0.018** |

---

> ### Author Response · Authors · 2024-11-20
> **Table 12-13, Table for Computational Complexity**
>
> |    GAMMA Dataset                      |                     | Glaucoma 3-class |                   |                   |
> | ------------------------ | ------------------- | ---------------- | ----------------- | ----------------- |
> |                          |                     | ACC              | AUROC             | AUPRC             |
> | Multimodal ，Single Task | Eyemost             | 0.860±0.017      | 0.910 ± 0.018     | 0.851 ± 0.022     |
> |                          | Eyestar             | 0.854±0.029      | 0.906 ± 0.022     | 0.841 ± 0.032     |
> |                          | M4oE without MSOE   | 0.862±0.018      | 0.912 ± 0.026     | 0.854 ± 0.025     |
> |                          | M4oE                | 0.876±0.025      | 0.927 ± 0.041     | 0.865 ± 0.026     |
> | Multimodal Multi task    | Eyemost             | 0.865±0.016      | 0.921 ± 0.022     | 0.858 ± 0.027     |
> | (medical ai)             | Eyemost+MToE        | 0.872±0.014      | 0.924 ± 0.014     | 0.861 ± 0.008     |
> |                          | Eyestar             | 0.859±0.022      | 0.899 ± 0.032     | 0.845 ± 0.033     |
> |                          | Eyestar+MToE        | 0.875±0.013      | 0.926 ± 0.008     | 0.865 ± 0.011     |
> | Multimodal Multi-task    | EVIF                | 0.887±0.017      | 0.936 ± 0.018     | 0.884 ± 0.016     |
> | (natural domain)         | Fuller              | 0.878±0.021      | 0.927 ± 0.019     | 0.877 ± 0.022     |
> |                          | AIDE                | 0.885±0.029      | 0.933 ± 0.025     | 0.881 ± 0.030     |
> |                          | MModN               | 0.892±0.016      | 0.943 ± 0.013     | 0.885 ± 0.014     |
> |                          | Multimodal Soft Moe | 0.871±0.029      | 0.918±0.027       | 0.863±0.025       |
> |                          | AdaMV-MoE           | 0.874±0.028      | 0.923±0.026       | 0.860±0.021       |
> |                          | Fuse-MoE            | 0.886±0.020      | 0.934±0.018       | 0.883±0.019       |
> |                          | Ours without MTOE   | 0.881±0.022      | 0.93 ± 0.039      | 0.879 ± 0.026     |
> |                          | Ours                | **0.904±0.017**  | **0.952 ± 0.015** | **0.895 ± 0.018** |
>
>
>
>
>
>
>
> | Multimodal MIMIC EXPERIMENTS |                             |                             |                      |                      |
> | ---------------------------- | --------------------------- | --------------------------- | -------------------- | -------------------- |
> |                              | In-hospital Mortality AUROC | In-hospital Mortality AUPRC | Length of Stay AUROC | Length of Stay AUPRC |
> | HAIM                         | 0.8091±0.054                | 0.4689±0.063                | 0.8174±0.035         | 0.7598±0.047         |
> | EVIF                         | 0.8179±0.046                | 0.5194±0.055                | 0.8156±0.037         | 0.7615±0.041         |
> | MModN                        | 0.8143±0.031                | 0.5225±0.039                | 0.8242±0.032         | 0.7777±0.028         |
> | Fuse-MoE                     | **0.8331±0.033**            | **0.5421±0.034**            | 0.8325±0.026         | 0.7844±0.024         |
> | Ours                         | *0.8310±0.039*        | *0.5368±0.041*         | **0.8561±0.025**     | **0.7891±0.037**     |
>
>
>
>
> | Complexity           | mirai  | asymmirai | EVIF   | Fuller | AIDE   | Ours | Fuse-MoE |
> | -------------------- | ------ | --------- | ------ | ------ | ------ | ---- | -------- |
> | inference time/batch | 0.37   | 0.38      | 0.5    | 0.35   | 0.36   | 0.47 | 0.44     |
> | training time/batch  | 1.15   | 1.01      | 1.56   | 0.98   | 1.12   | 1.51 | 1.32     |
> | params               | 120.4M | 205.4M    | 188.1M | 211.6M | 254.3M | 3.2B | 2.1B     |

---

> ### Author Response · Authors · 2024-11-25
>
> Dear Reviewer P7K2,
>
> Thank you very much for your valuable feedback. We have provided comprehensive point-by-point responses to address each of your concerns and questions. As we approach the end of the discussion period, we would greatly appreciate your feedback on whether our responses have adequately addressed your points. We remain available to provide any additional clarification you may need.
>
> Best regards,
>
> Submission 11923 Authors

---

> ### Author Response · Authors · 2024-11-28
>
> Dear Reviewer P7K2,
>
> Thanks very much for your time and valuable comments. We understand you're very busy. As the window for responding and paper revision is closing, would you mind checking our response and confirming whether you have any further questions? We are happy to provide answers and revisions to your additional questions. Many thanks!
>
> Best Regards,
>
> Authors of Submission 11923

---

> ### Author Response · Authors · 2024-12-03
> **Gentle Reminder**
>
> Dear P7K2,
>
> Thank you again for your valuable review! We’ve provided detailed responses and results to answer your questions. As we are near the end of the discussion period, we would greatly appreciate it if you could review our responses and share any additional feedback you may have. We remain available to address any further questions or concerns. Thanks a lot!
>
> Best Regards,
>
> Authors of Submission 11923

---

### Author Response · Authors · 2024-11-21
**Global Response for All Reviewers**

**For all reviewers**

We thank all reviewers for their valuable suggestions and constructive feedbacks, which have significantly contributed to the improvement of our paper.

Our work introduces a novel model for dynamic multimodal fusion and modeling task-modality dependence for practical multimodal multi-task problem in the medical imaging domain. We are glad that all reviewers have recognized the originality of our work and the importance of the problem, as well as the contributions of our work with the solid performance and soundness of our approach, as noted by reviewers P7K2, xATa, 7L3d, dm8E.

**Main Changes**

We have revised the paper to carefully address all the reviewers’ questions. Our updated manuscript has been uploaded to the OpenReview system. As suggested by reviewers,10 additional result tables are included in the updated appendix.

1. **More Baselines, Non-Imaging Modalities and Metrics**: We've included detailed experimental results that compare our approach with 3 more baselines recently published. We also illustrate the generalization strength of our method using an additional dataset MIMIC-CXR including multi-modalities: X-ray images+Radiology reports+EHR. We additionally report the AUROC and AUPRC metrics along with standard deviation across cross-validation. This addresses questions from reviewers P7K2, 7L3d, dm8E.

2. **Computational Complexity**: We have reported the computational cost and analysis to address questions from reviewers P7K2, xATa, 7L3d.

3. **More Ablation Study and Analysis**: The revision have featured more ablation study results (reviewer xATa) and analysis including but not limited to modality uniqueness (reviewer 7L3d) and subgroup performance (reviewer dm8E).

4. **Related work and More Clarifications**: We have included valuable literature suggestions from reviewers xATa and dm8E with more discussion added in the related work section. We also added a clinical interpretation to help explain our results.

---

> ### Public Comment · ~Shiwen_Zhnag1 · 2024-11-28
> **How to incorporate Flex-MoE**
>
> Thanks authors for the impressive work! This work is very interesting. I had a question for  the future work on missing modalities. In page 11 Line 540, authors introduce an embedding bank technique by Flex-MoE [1]. How to adapt that technique to the MoSE, MoTe framework? It would also be valuable if authors can evaluate the performance of Flex-MoE on the tasks given in the paper as a future work.  Thanks again!
>
> [1]. Yun, Sukwon, et al. "Flex-MoE: Modeling Arbitrary Modality Combination via the Flexible Mixture-of-Experts." The Thirty-eighth Annual Conference on Neural Information Processing Systems.

---

> > ### Author Response · Authors · 2024-12-03
> >
> > Dear Shiwen,
> >
> >   Thank you for your good comments! While missing modality is beyond the scope of our current paper, it's an important direction we plan to explore in future work, particularly in comparison with Flex-MoE's approach.
> >
> >    A lot of current works for missing modality use padded or imputed inputs. Flex-MoE introduced an innovative approach by creating learnable embeddings for each possible missing modality combination (e.g., for modalities A and B, handling cases where A is missing or B is missing). These embeddings are positioned after modality-specific encoders rather than at the input level or post-MoE. This is what they mention by “training each encoder solely with observed samples”, because the missing data does not go through the encoders in the form of zeros or other imputed values. This approach could be readily adapted to our modality-specific encoders.
> >
> >   For the exact pytorch implementation, the authors of Flex-MoE defined an encoder dictionary for each modality and a learnable missing embedding bank. Taking imaging modality for example, it consists of a torch.nn.Sequential module that combines Custom 3D CNN and PatchEmbeddings. The original code for learnable missing embedding as follows:
> >
> >         if num_modalities > 1:
> >                   missing_embeds = torch.nn.Parameter(torch.randn((2**num_modalities)-1, args.n_full_modalities, args.num_patches, args.hidden_dim, dtype=torch.float, device=device), requires_grad=True)
> >                   params += [missing_embeds] #this is to help make the parameters trainable
> >
> > The core code to replace missing modality embedding with this learnable embedding:
> >
> >         for i, (modality, samples) in enumerate(batch_samples.items()):
> >             mask = batch_observed[:, modality_dict[modality]]
> >             encoded_samples = torch.zeros((samples.shape[0], args.num_patches, args.hidden_dim)).to(device)
> >             if mask.sum() > 0:
> >                 encoded_samples[mask] = encoder_dict[modality](samples[mask])
> >             if (~mask).sum() > 0:
> >                 encoded_samples[~mask] = missing_embeds[batch_mcs[~mask], modality_dict[modality]]
> >             fusion_input.append(encoded_samples)
> >
> >
> >   Finally, in our latest response to reviewer dm8E, we include extreme cases where modality suffers from a large amount of noise. We simulate this by applying very strong artificial noise to corrupt one of the modalities randomly. Our model shows good performance under these extreme cases and allocates dynamic modality utilization based on information quality. This shows that our model can deal with very noisy data and potentially can also be good against missingness. Please refer to Table 19: Edge Case (Modality Corruption Analysis) on RSNA Dataset for more details.
> >
> >   We hope this can be helpful to your question, and we look forward to keep exploring this direction in our future work. Thank you!
> >
> > Best Regards,
> >
> > Authors of Submission 11923

---

### Meta-Review · Area_Chair_BoPP · 2024-12-15

**Metareview:**

The submission proposes strategies for multi-modal learning based on specific mixture-of-experts models for patient level and task level fusion.  The approach uses a conditional mutual information regularization between modalities and experts.  It is an interesting contribution that marries multimodal learning with MoE models in a sensible framework, which is an interesting contribution.  The work is contemporary with Flex-MoE (NeurIPS 2024), and is well timed given unique challenges in multimodal learning, and the use of highly successful MoE approaches in LLMs.  3 of 4 reviewers were weakly positive about the submission, while one felt that it was below the threshold.  On the balance, the submission is interesting and of good quality.

**Additional Comments On Reviewer Discussion:**

The reviewer who was most negative did not respond to the extensive rebuttal from the authors.  Other reviewers who were active in the discussion increased their score as the result of the rebuttal.  The authors were very active in providing additional information to reviewer requests.  Timing results are important, but should not be confused with computational complexity (usually expressed in big-O), as appears to be the case in the author response.  The authors did not actually answer the reviewers questions on this point, but did provide timing results indicating that the overheads could be OK in practice.

---

### Decision · Program_Chairs · 2025-01-22

Accept (Poster)